# Time-resolved dynamic computational modeling of human EEG recordings reveals gradients of generative mechanisms for the MMN response

Arnaud Poublan-Couzardot[1], Françoise Lecaignard[1], Enrico Fucci[2], Richard J. Davidson[3,4,5,6], Jérémie Mattout[1], Antoine Lutz[1‡], Oussama Abdoun[1‡*]

**1** Cente de Recherche en Neurosciences de Lyon (CRNL), CNRS UMRS5292, INSERM U1028, Université Claude Bernard Lyon 1, Bron, France, **2** Institute for Globally Distributed Open Research and Education (IGDORE), Sweden, **3** Center for Healthy Minds, University of Wisconsin, Madison, Wisconsin, United States of America, **4** Department of Psychology, University of Wisconsin, Madison, Wisconsin, United States of America, **5** Waisman Laboratory for Brain Imaging and Behavior, University of Wisconsin, Madison, Wisconsin, United States of America, **6** Department of Psychiatry, University of Wisconsin, Madison, Wisconsin, United States of America

‡ These authors are joint senior authors on this work.
* oussama.abdoun@pm.me

**Data Availability Statement:** The source code and data used to produce the modeling results

## Abstract

Despite attempts to unify the different theoretical accounts of the mismatch negativity (MMN), there is still an ongoing debate on the neurophysiological mechanisms underlying this complex brain response. On one hand, neuronal adaptation to recurrent stimuli is able to explain many of the observed properties of the MMN, such as its sensitivity to controlled experimental parameters. On the other hand, several modeling studies reported evidence in favor of Bayesian learning models for explaining the trial-to-trial dynamics of the human MMN. However, direct comparisons of these two main hypotheses are scarce, and previous modeling studies suffered from methodological limitations. Based on reports indicating spatial and temporal dissociation of physiological mechanisms within the timecourse of mismatch responses in animals, we hypothesized that different computational models would best fit different temporal phases of the human MMN. Using electroencephalographic data from two independent studies of a simple auditory oddball task (n = 82), we compared adaptation and Bayesian learning models' ability to explain the sequential dynamics of auditory deviance detection in a time-resolved fashion. We first ran simulations to evaluate the capacity of our design to dissociate the tested models and found that they were sufficiently distinguishable above a certain level of signal-to-noise ratio (SNR). In subjects with a sufficient SNR, our time-resolved approach revealed a temporal dissociation between the two model families, with high evidence for adaptation during the early MMN window (from 90 to 150-190 ms post-stimulus depending on the dataset) and for Bayesian learning later in time (170-180 ms or 200-220ms). In addition, Bayesian model averaging of fixed-parameter models within the adaptation family revealed a gradient of adaptation rates, resembling the anatomical gradient in the auditory cortical hierarchy reported in animal studies.

presented in this manuscript are available from OSF at: https://osf.io/dxj5p/.

**Funding:** This work was supported by grants from the National Center for Complementary and Integrative Health (NCCIH) to RJD (P01AT004952); a core grant to the Waisman Center from the National Institute of Child Health and Human Development (NICHD) (HD003352); a European Research Council (ERC) Consolidator grant to AL (617739-BRAINandMINDFULNESS); a Mind & Life Europe Varela Award grant to OA (2017-EVarela-Abdoun); and supported by the LABEX CORTEX (ANR-11-LABX-0042) of Université de Lyon, within the program "Investissements d'Avenir" (ANR-11-IDEX-0007) operated by the French National Research Agency (ANR). The funders had no role in study design, data collection and analysis, decision to publish, or preparation of the manuscript.

**Competing interests:** The authors have declared that no competing interests exist.

## Author summary

The ability to detect and adapt to changes in the environment is an essential feature for survival of living beings. Two main theories have been proposed to explain how the brain performs such an automatic task in the auditory domain. The first one, adaptation, emphasizes the ability of auditory cortical and sub-cortical neurons to attenuate their response to repeated stimuli, which renders the brain more sensitive to deviations from expected sensory inputs. The second one, Bayesian learning, further involves higher-level cortical regions which would update their predictions about incoming stimuli, depending on their performance at predicting previous ones. These two views may not be mutually exclusive, but few experimental works compared them directly. We used computational models inspired from both accounts to assess which view may provide a better fit of two independent electrophysiological datasets from similar auditory experiments. Evidence from a large sample of 82 human subjects provided a complex picture, with adaptation processes seemingly dominating the early phase of auditory brain response, and Bayesian learning processes appearing later on. Our results converge with other recent works in animals and points to the necessary reconciliation of those two theories for a better understanding of auditory perception and statistical learning.

## Introduction

### The MMN, a widely used tool in neuroscientific research

The auditory mismatch negativity (MMN) is a well-investigated electrophysiological marker of automatic neural detection of changes occurring in the auditory environment [1] and is typically induced using an oddball paradigm where sequences of frequent standard tones are presented interspersed with rare deviant tones. In humans, it is recorded through electroencephalography (EEG) or magnetoencephalography (MEG) and is defined as the difference between the deviant and standard event-related potentials (ERPs) [2]. Multifarious deviance features have been shown to induce a MMN response, including pitch, duration, intensity, spatial location, etc. The MMN is even observed in response to violations of abstract regularities that bear no relationship with the physical features of the stimuli [3].

The MMN is a negative wave generally peaking over fronto-central scalp electrodes between 100 and 250 milliseconds, with variations depending on the experimental design including the dimension of the deviant feature [1]. Discovered more than 40 years ago [4], it was investigated in research fields as broad as neurolinguistics [5], attention [6], altered states of consciousness [7, 8], aging [9], psychiatry [10, 11], emotions [12], psychedelics [13] or meditation [14]. However, despite its use in clinical research or as an experimental measure for many cognitive functions [15], its generative mechanisms are still widely debated.

### An old debate around the generative mechanisms of the MMN

Recent reviews described in details the different mechanistic hypotheses for the mismatch response [16, 17]. The MMN was originally viewed as a marker of sensory memory by its discoverers [18], but such *memory-based* account quickly found its limits in the face of the existence of MMN responses to abstract deviations [3] and was refined under the *model-adjustment hypothesis* [19, 20]. In this view the main mechanism behind the MMN is not only the detection of changes in some physical features of a stimulus, but more generally the

detection, representation and updating of regularities in the auditory stream. Being more flexible, the model adjustment account can explain the MMN to most forms of deviance, from omission responses to violations of abstract rules.

Another prevailing explanation for the MMN is the *adaptation* hypothesis, which rests upon electrophysiological observations of stimulus-specific adaptation (SSA) in auditory brain regions at the origin of deviance detection [21–23]. Indeed, midbrain [24], thalamic [25] and cortical [26–29] auditory neurons tuned to frequently repeated tones adapt faster than neurons tuned to rare deviant tones. This differential adaptation at the neuronal level would explain the lower amplitude of the scalp ERP-component N1 in response to frequent, compared to deviant tones, resulting in the observation of the negative MMN deflection. [30–32]. If critics were raised concerning the explanatory power of the adaptation hypothesis and its relation to the MMN [33–35], most of them were fairly answered with simulations works [30, 31, 36, 37] and detailed reviews [22, 38]. As May et al. [22] highlighted, memory-based and adaptation accounts fall in two different categories: cognitive and physiological, respectively, which are not mutually exclusive explanations of the MMN but justify further efforts in the development of a more encompassing theory of deviance detection.

Bayesian inference, and its implementation in the brain as *predictive coding*, has been used extensively in the last two decades as a general model of perception [39–43], with particular emphasis on audition [44–46]. In predictive coding, the MMN is seen as an electrophysiological marker of the discrepancy between top-down predictions based on repeated stimuli (i.e. standards) and bottom-up unexpected sensory inputs (i.e. deviants): a prediction error [40]. Propagated through forward connections, prediction errors would trigger the updating and optimization of an internal model of sensory causes at higher cortical areas, in agreement with the model-adjustment hypothesis. Adjusted predictions would then be sent backward to explain away or suppress prediction errors at lower cortical areas, providing a plausible mechanism for repetition suppression. In this framework, suppression of the MMN when deviant tones are repeated and thus become standards, as in the roving standards paradigm [47], is interpreted as a minimization of prediction errors, the internal model having learnt a new statistical regularity. Additionally, hierarchical minimization of prediction errors would also be weighted by the relative precision of the predictions and sensory inputs, accounting for effects of standard variability and deviant probability on MMN amplitude [48]. Physiologically, changes in local, intrinsic connectivity of prediction error units would mediate this weighting process in auditory cortices via modifications of synaptic gains, reminiscent of the adaptation account. Using Dynamic Causal Modelling (DCM) on human EEG data, Garrido and colleagues showed that model-adjustment or adaptation hypotheses alone are not sufficient to explain changes in extrinsic and intrinsic connectivity specific to the MMN in both classical [49–52] and roving [53] auditory oddball paradigms. Converging with previous attempts [54], they proposed that both hypotheses are necessary to fully account for N1 and MMN brain responses, and are reconcilable under predictive coding [55]. Empirical support for this integrated account has come from the previously mentioned DCM studies on EEG [49–53] and MEG data [56, 57] as well as EEG-MEG fusion [58] and electrocortigraphy (ECoG) recordings [57]; but also from more classical ERP and sources reconstruction analyses using refined paradigms with EEG and MEG [59] or ECoG [60], to reveal the hierarchical organization of deviance detection processes in auditory, associative and frontal cortices.

In parallel to the above MMN studies in humans, an extensive field of research has developed on deviance detection in the animal brain. The possibility to use more invasive methods on cats and rodents allows to gain additional insights on the neural mechanisms of deviance processing.

## Insights from animal studies

Animal studies of deviance detection have focused on the neuronal dynamics and location of such effects. In this context, SSA has been identified as the core neuronal mechanism at the cellular level [26, 27], the MMN being its macroscopic correlate and reflecting the activity of multiple neuronal populations summed together [21]. However, theoretical and methodological issues were encountered while trying to integrate SSA and MMN under the same deviance detection mechanism [21, 61]. More recently, following human research, results in animal studies have been reinterpreted under the predictive coding framework with convincing explanatory power [16, 62]. Building on single neuron recordings and new paradigms allowing to disentangle mere repetition suppression (adaptation mechanism) from prediction error [16], researchers were able to demonstrate empirically the hypothesized hierarchical organization of predictive coding in the auditory system at the neuronal level [63]. If neuronal mismatch activity was predominantly explained by repetition suppression, a significant part could also be accounted for by "pure" deviance detection (prediction error). Notably, such prediction error–like activity was observed at multiple levels of the neural hierarchy, arising gradually from the inferior colliculus in the midbrain up to thalamic and cortical areas [63]. In addition, the neurons in the non-lemniscal divisions of the recorded brain areas systematically showed higher prediction error–like activity than their lemniscal counterparts at each level of the auditory hierarchy. This empirical finding is consistent with the known anatomical and functional connectivity of the non-lemniscal pathway. Carabajal and Malmierca (2018) have argued that the fact that broadly tuned non-lemniscal neurons receive the bulk of descending cortical signals, as well as inputs from the neighbouring, sharply tuned and tonotopically organized lemniscal neurons, make them a good candidate for hierarchical predictive coding [16]. Interestingly, in a subsequent study, the same authors tested again this dichotomy at frontal areas, confirming the existence of pure, unconfounded by repetition suppression, mismatch responses in the prefrontal cortex (PFC) of anesthetized rats [64]. Those responses differ in multiple aspects from auditory cortex responses, adding further evidence for the hypothesis of divergent but complementary roles of frontal and sensory areas in predictive coding. Indeed, responses at prefrontal areas were mostly explained by a prediction error index and occurred later than auditory cortex responses dominated by repetition suppression mechanisms.

## Insights from computational modeling

Computational modeling has been of great help to try to better understand MMN generation mechanisms under adaptation and predictive coding accounts. Independent modeling efforts have designed models of adaptation at the neuronal [30, 65, 66] and cortical level [37, 67], while a predictive coding model was implemented with a neuronal architecture simulating a thalamo-cortical network [68]. Additionally, more phenomenological models linked electrophysiological mismatch responses to hierarchical Bayesian inference in auditory [69, 70]. However, if these independent models successfully accounted for critical SSA or MMN features, few studies directly compared models derived from divergent MMN theories. A first attempt found a Bayesian learning model, consistent with predictive coding accounts, to outperform classical "stimulus change" (or change detection) models at predicting somatosensory mismatch EEG responses at the source level [71]. Recent work extended the model to include different neural surprise signatures and could show interesting spatial and cortical dynamics of the somatosensory mismatch response. Similar computational models were also applied to the auditory MMN in fused EEG-MEG [72] and ECoG [73] measurements, and to the visual MMN in EEG [74], confirming the previous results and designing new Bayesian learning models of deviance detection. Finally, Grundei et al. very recently showed evidence for

Bayesian learning against classical change detection to explain cross-modal EEG mismatch responses (auditory, somatosensory and visual) in humans [75].

Only one study ([76]) directly compared all major MMN theories: change detection, adaptation and Bayesian inference, with the latter nuanced in three distinct models reflecting different views on which aspect of statistical learning is reflected by the MMN (prediction error, Shannon surprise or model adjustment). In their 2013 modelling study of auditory MMN responses elicited by a roving oddball paradigm [76], Lieder and colleagues found substantially higher evidence for the set of Bayesian models compared to the set of alternative models (adaptation and change detection models). However, this result hinged on the model comparison being carried out at the level of families aggregating multiple models, and no conclusion could be drawn at the level of individual models, possibly due to an insufficient sample size (8 participants). Critically, the analysis was performed on a time-averaged signal, rendering it blind to any temporal differences in MMN generation processes.

### The current study

Building on the strengths of previous modeling studies and addressing some of their methodological shortcomings, we used two independent EEG datasets of 28 and 54 subjects (final sample size after preprocessing) recorded from similar auditory oddball paradigms to compare five model families, each encoding a prevailing MMN theory (change detection, adaptation, prediction error, Shannon surprise and model adjustment). We employed a time-resolved dynamic modeling approach to test a putative temporal dissociation in the computational mechanisms explaining the MMN. As a first validity check and critical improvement from former studies, we ran confusion analyses on simulated data with varying signal-to-noise ratio (SNR). Under our experimental design and models set, model recovery on synthetic data was highly dependent on SNR and allowed us to estimate the reliability of model comparison results obtained from empirical data. Bayesian model fitting and comparison revealed a clear distinction in model families' dominance between early and late windows of the MMN, that could not be explained by confusion between the models. In addition, we could identify a temporal gradient in the rate of adaptation models. We discuss these computational results obtained from human scalp responses in light of recent electrophysiological results in the animal literature.

## Materials and methods

### Ethics statement

For the present trial-by-trial modeling study, we used datasets from two published EEG experiments investigating the impact of meditation states and meditation expertise on the amplitude of the MMN evoked potential [14], [77]. The research protocol for Study 1 was approved by the UW-Madison Health Sciences Internal Review Board. For Study 2, ethical approval was obtained from the appropriate regional ethics committee on human research (CPP Sud-Est IV, 2015-A01472–47). All participants from both studies provided written informed consent before the beginning of the experiments.

The original experiments have been extensively described in the corresponding articles; below we report only the materials and methods relevant to the current study. In particular, only the data from the control conditions (defined below) of the two original experiments have been used for modeling.

To ensure the quality of the modeled data and thus of the results derived from it, participants with insufficient data after preprocessing or lacking a detectable MMN based on objective criteria were rejected.

The last part of the methods section describes in details our modeling approach. We introduce the specific framework used and detail the computational implementations of the prevailing accounts offered as generative models of the MMN response. The parameters of the model fitting procedure are provided for replication purposes. The approaches we implemented for inference based on Bayesian model comparison as well as model recovery analysis are described with their underlying rationale. A last section concerns the measure used to evaluate the absolute goodness-of-fit of the selected models.

## Study 1

**Participants.**   Data from 31 healthy participants (25 males, mean age of 42.9 ± 10.4 years old) were collected.

**Experimental design.**   Subjects underwent a passive auditory oddball task [78] consisting of the variable repetition of a standard tone (1000 Hz; 60 ms duration; 10 ms rise and fall; 80 dB SPL) followed by the presentation of a frequency deviant tone (1200 Hz; 60 ms duration; 10 ms rise and fall; 80 dB SPL). Each block of the task contained 80% standard tones (n = 200) and 20% deviant tones (n = 50) with a variable inter-stimulus interval (ISI) of 800—1200 ms. Each subject underwent three blocks of three conditions each: two meditative practices and one control condition. During the control condition, subjects were instructed to read a newspaper and ignore the auditory stimulation. We refer the interested reader to the mentioned article [14] for details about the meditation instructions. The order of conditions within a block was randomized but all subjects had the same order, with the control condition being the first condition in the first block and then the third condition in the second and third block. For the present article we will only look at data from the control condition, consisting in 750 stimuli (150 deviants) per participant before data rejection.

EEG data was collected with a 128-channel Geodesic Sensor Net (Electrical Geodesics, Eugene, OR), sampled at 500 Hz, and referenced to the vertex (Cz).

**EEG data preprocessing.**   Data were pre-processed using the EEGLAB software [79] on Matlab (The Mathworks Inc.). Notch filtering was applied at 60 Hz to remove line noise, followed by the application of a band-pass filter between 0.5 and 100 Hz. Raw data were manually cleared of large movement-related artefacts and bad channels were identified and interpolated. Independent Component Analysis (ICA) was applied to the raw data of each participant (only on the non-interpolated channels) using runica (Infomax) algorithm [80] to identify and remove artefacts caused by eye blinks and saccades, as well as cardiac and muscular activity. After ICA correction, data were re-referenced offline to the average of both mastoids, a non-causal band-pass 1—60 Hz digital filter was applied, and two-second epochs centred on stimulus onset (-1 to 1 s) were extracted. Epochs were visually inspected and the ones still presenting muscular or eye movement artefacts were marked for rejection. Finally, the epoched data were baseline-corrected by subtracting the mean value of the signal during the 100 ms window preceding stimulus presentation.

We focused the present analysis on the same region of interest (ROI) as Fucci and colleagues [14], a frontal area comprising 12 electrodes selected based on the previous literature (including Fz). This ROI can be visualized in S1 Fig. The data fed to the models was always the average over these ROI electrodes.

## Study 2

**Participants.**   The second dataset used for modeling comes from a subsequent, replication study by the same authors [77]. 66 participants were included in this study (30 females, mean age of 52 ± 7.7 years old).

**Experimental design.**    The experiment consisted in a passive auditory oddball task with sequences of standard tones of variable length (880Hz; 80ms duration; 10ms rise and fall) followed by the presentation of a frequency deviant (988Hz; 20% of all auditory stimuli). The stimuli were presented binaurally with a fixed ISI of 500ms. A threat induction procedure [81] was added to the experiment. Participants were exposed to two experimental conditions while passively listening to the tones: "threat" periods during which they were informed they could receive an electrical shock on the wrist (whose amplitude was calibrated beforehand, based on the participant pain threshold), and "safe" periods where they knew no shocks would be delivered. Safe and threat periods were alternating randomly every 70 stimuli (35 seconds) and participants were cued to the change. This threat induction oddball paradigm was split into three blocks where subjects were either asked to: watch a silent movie (control condition), practice a focused attention meditation or practice an open presence meditation. The order of the blocks were randomized between participants. 224 standards and 56 deviants were presented per block and per condition (4 safe and 4 threat periods per block). The full paradigm was repeated in a second session two hours later. For the purpose of the current article we will focus on EEG data from the control condition (watching a silent movie) and safe periods (no threat of electrical shock) only, leading to 560 stimuli (112 deviants) per participant before data rejection. EEG data was collected in a shielded Faraday chamber with a 64 electrodes Biosemi ActiveTwo system at a 512 Hz sampling frequency, offset was maintained within 50mV (± 25) as recommended in the Biosemi guidelines.

**EEG data preprocessing.**    Pre-processing was done using EEGLAB [79] and in-house Matlab scripts (Matlab version R2017a, The Mathworks Inc). EEG data was downsampled to 250 Hz, re-referenced to the average of the mastoids and visually inspected, noisy channels (deficient electrodes, long-lasting muscular artefacts, etc) were marked and removed temporarily. For every subject, and separately for the two sessions, Independent Component Analysis (ICA, with runica algorithm) was applied on 1–20 Hz filtered data (EEGLAB eegfiltnew function, Hamming windowed sinc FIR filter of order 826) to improve ICA decomposition. ICA weights hence obtained were applied to the unfiltered data and ICA components reflecting blinks and eye movements were manually selected and rejected. Subsequently a 2 Hz high-pass filter (Hamming windowed sinc FIR filter of order 414) was applied to remove low-frequency drifts most likely caused by stress-related perspiration induced by the threat of electrical shocks. Previously marked bad channels were interpolated using EEGLAB spherical splines interpolation and a line noise removal algorithm (Cleanline, linefreqs = 50, bandwith = 2, tau = 100, winsize = 4, winstep = 1) was applied.

Cleaned data were epoched between -200ms and 500ms around auditory tones. The baseline was chosen as the 100 ms period before stimulus onset, as recommended in most of the MMN literature, and removed for subsequent analyses. Finally epoched data were low-pass filtered at 60 Hz (order 56), a ± 75 mV rejection threshold was applied and epochs marked for rejection visually inspected. For this article where only data from the control state (watching a video) were analysed, the recording for one subject was excluded due to an excessive amount of high-frequency noise and two others because they ended up with less than 35 remaining good epochs per condition to compute the MMN (deviant and standard before deviant). This led to 63 participants remaining for Study 2 after EEG preprocessing.

For subsequent analyses we selected the same frontal ROI as [77] (S1 Fig), comprising only 5 electrodes but similarly located as the ones in Study 1 when taking into account the difference in electrodes layout (EEG system with 64 electrodes for Dataset 2 instead of 128 for Dataset 1). We always modeled the average over these 5 electrodes ROI.

## Validation of individual MMN

The MMN is a rather robust component observable in virtually all human participants, but in some occasions it might fail to be detected even in healthy participants with intact sound discrimination [82, 83]. We used a simple automatic procedure to identify those participants that did not show a detectable MMN. For each participant, a Gaussian function was fitted on the standards minus deviants ERPs difference, averaged over all selected ROI electrodes, by optimizing two free parameters—the offset level and the standard deviation—as well as two constrained parameters: the temporal location of the peak forced between 100 and 200ms and the amplitude of the peak, restricted to negative values. If the optimizer converged on a zero amplitude solution, it was interpreted as failing to detect a negative deflection in the signal and the participant was excluded.

The results of the Gaussian fit on the ERP procedure are represented in S2 and S3 Figs. As a result of the procedure, 3 participants out of the 31 in study 1 were excluded, as well as 9 out of 63 in study 2.

## Signal-to-noise ratio

We define the signal-to-noise ratio (SNR) of EEG as the ratio between signal power and noise power:

$$\text{SNR} = \frac{P_{signal}}{P_{noise}} = \frac{E[S^2]}{E[N^2]} = \frac{E[S]^2 + Var[S]}{E[N]^2 + Var[N]}$$

Assuming that the noise $N$ has a mean of 0 (i.e. $E[N] = 0$) and that most of the signal variance can be attributed to noise ($Var[N] \approx Var[S]$):

$$\text{SNR} \approx \frac{\mu^2 + \sigma^2}{\sigma^2} = 1 + \frac{\mu^2}{\sigma^2} \tag{1}$$

where $\mu$ and $\sigma^2$ denote the mean and variance of the signal, respectively. Therefore the SNR can be written as a simple function of the amplitude of the averaged ERP and the intertrial variance.

## Modeling

**Modeling framework.** The goal of this study was to compare alternative accounts of MMN generation, namely adaptation and Bayesian learning. Importantly, those accounts are based on the temporal dependency to previously heard tones, a feature that is difficult to explore with traditional average-based ERP analyses. Thus we used dynamic modeling to explore trial-by-trial fluctuations of deviance detection processes. Static models were also designed to serve as controls. To model MMN responses in a dynamic way we used variational Bayes inference as implemented in the VBA toolbox [84]. This framework uses non-linear state-space models to predict a *response y* (or measurement, e.g. EEG scalp activity) to experimental *inputs u* (e.g. auditory tones), given internal (hidden) states *x* that are not directly observable. Such models are composed of two functions working together. First an **evolution function *f*** describes how internal *hidden states x* evolve over time (here represented by trial index *i*):

$$\boldsymbol{x}_{i+1} = f(\boldsymbol{x}_i, u_i, \boldsymbol{\theta}) \tag{2}$$

where $\boldsymbol{\theta}$ is a vector of *evolution parameters* and $u_i$ codes the identity of trial $i$ (standard tones: $u = 0$, deviant tones: $u = 1$).

Second we need to define an **observation function $g$** that computes the *predicted response* generated by hidden states $x$. The predicted response is connected to neural activity through a simple linear function:

$$y_i = \phi_0 + \phi_1 g(x_i, u_i, \lambda) + \epsilon_i \tag{3}$$

where $\phi_0$ and $\phi_1$ are (fitted) *observation parameters* and $\epsilon$ the Gaussian measurement noise which is always modeled but will be omitted in later description of the models for the sake of simplicity. $\lambda = (\lambda_0, \lambda_1)$ is a model-dependent pair of offset and scale factor that normalize the observation function to a similar range across models. Unlike $\phi$, $\lambda$ is not fitted but estimated for each model using synthetic data, as such they are not model parameters *per se*. These normalization constants allow the parameters $\phi$ to share the same interpretation across models. In practice, we set $\lambda$ so that $\phi_0$ can be interpreted as the mean response to standard tones ($\bar{y}_{u=0}$), and $\phi_1$ as the mean deviant minus standard difference potential ($\bar{y}_{u=1} - \bar{y}_{u=0}$).

Model inversion over all EEG trials was performed independently for every model, subject and 10ms averaged time window ranging from 45 to 255ms post–stimulus. It issued inversion diagnostics, model's posterior parameters estimation and goodness–of–fit measures which will be described later. We will now present in more detail the generative models used to formalize the MMN hypotheses at stake.

**Generative models of brain responses.**   We will first describe the two static models used as controls, namely the *null model* and the *deviant detection*. By definition, static models have no evolution function.

The **Null model** predicts a constant response to any auditory stimulus.

$$g(u_i) = 0 \tag{4}$$

**Deviant detection (DD)** model predicts the MMN to be a response to deviant stimuli only:

$$g(u_i) = u_i \tag{5}$$

**Adaptation** models are based on the neuronal adaptation hypothesis [22] which sees the auditory MMN as resulting from a difference in adaptation of frequency-specific cortical neurons, a process often called stimulus specific adaptation (SSA). Neurons tuned to the frequency of standard tones adapt quickly with recurrent repetitions thus decreasing their responsiveness, while neurons responding preferentially to the frequency of deviant tones presented rarely will adapt less and therefore produce a stronger response on average than their standard counterparts. SSA has been observed throughout the entire auditory hierarchy, particularly in cortical areas [23]. It is commonly and adequately modeled computationally as a process of exponential decay [73, 85, 86]. For the sake of simplicity we will abbreviate adaptation models to *SSA*, from the supposed underlying electrophysiological mechanism, yet one has to keep in mind that we do not model a biologically plausible implementation of SSA (see [30, 37, 65, 67] for such works) but only its phenomenological approximation as exponential decay.

In the implementation of adaptation dynamics two hidden states ($x_0, x_1$) represent neuronal responsiveness to standards and deviants, respectively. Adaptation and recovery are each modeled with exponential functions. Neural activity in response to the presented tones is then

supposed to be proportional to the responsiveness of tone-specific neuronal populations:

$$f(x_i, u_i) : \begin{cases} \text{if } u_i = 0 \text{ (standard)} & \begin{cases} x_{0,i+1} = x_{0,i}K_a \\ x_{1,i+1} = 1 - (1 - x_{1,i})K_r \end{cases} \\ \text{if } u_i = 1 \text{ (deviant)} & \begin{cases} x_{0,i+1} = 1 - (1 - x_{0,i})K_r \\ x_{1,i+1} = x_{1,i}K_a \end{cases} \end{cases}$$

$$\text{with } K_a = e^{-1/\tau_a} \text{ and } K_r = e^{-1/\tau_r}$$

$$g(x_i, u_i, \lambda) = \lambda_0 + x_{u_i,i}\lambda_1$$

(6)

Single-cell recordings in the cat auditory cortex (A1) have revealed the existence of multiple timescales of adaptation ranging from less than a second to several tens of seconds with recovery rates about twice as high [27]. Based on this literature we implemented seven SSA models with different adaptation parameters $\tau_a \in [3, 10, 20, 30, 50, 100, 200]$. The recovery parameter $\tau_r$ was always set to be twice $\tau_a$, as informed by electrophysiological data on auditory neurons of anesthetized cats [27].

**Bayesian Learning (BL)** models are based on the predictive coding view of the MMN, they assume the brain learns the probability for a deviant to occur. We were inspired from computational models initially designed by Ostwald et al. [71], which treat the MMN as a measure of Bayesian surprise: the difference between prior and posterior distributions of beliefs on states of the world (here states being the category of tone stimulus: deviant or standard). Such models also implement an approximate learning of the statistical regularities of the environment by setting a "forgetting" constant. In their study, they showed that these models predict the somatosensory mismatch response better than control models that do not learn. This finding was replicated in the auditory modality combining EEG and MEG measurements [72].

A common evolution function models auditory tones as the outcome of a *Bernouilli process* of parameter $\mu \in [0, 1]$, the probability to have a deviant. The belief over $\mu$ follows a *Beta distribution*: $\mu \sim Beta(\alpha, \beta)$ with parameters $\alpha$ and $\beta$, tracking the number of deviants and standards that have been heard so far, respectively. The particularity of BL models is to compare prior and posterior beliefs over $\mu$. We thus assigned hidden states to $\alpha$ and $\beta$ parameters of the Beta distribution over $\mu$. $x_{1,i}$ and $x_{2,i}$ therefore define the **prior belief** over deviant's probability before having heard the current tone $u_i$, while $x_{1,i+1}$ and $x_{2,i+1}$ define the **posterior belief**. We can see that the main characteristic of these hidden states is to implement a memory of past inputs. However, to be biologically plausible and to take into account recent inputs history in a way that better represents ongoing dynamics, this memory should be finite and weigh recent events more than older ones. Following previous implementations [71, 72], we weighted the stimulus counts by a "forgetting" constant depending on a *temporal integration parameter $\tau_i$*: $K_t = e^{-1/\tau_t}$, which is hypothesized to account for different temporal integration windows. Hence with varying forgetting constants, this model can be seen as having more or less "memory" or, in a predictive coding scheme, as having more precise prior beliefs and associated predictions to make its perceptual decision. Thus it can "learn" if $\tau_t$ is sufficiently high ($K_t \to 1$, slow forgetting) or not if it is too low ($K_t \to 0$, fast forgetting). Based on our previous work in the auditory modality [72] we defined models for $\tau_t \in [5, 10, 20, 30, 50, 100]$.

$$f(x_i, u_i) : \begin{cases} x_{1,i+1} = u_i + x_{1,i}K_t \\ x_{2,i+1} = (1 - u_i) + x_{2,i}K_t \end{cases} \text{with } K_t = e^{-1/\tau_t}$$

(7)

Multiple hypotheses exist in the predictive processing literature about which internal mechanism is reflected by the MMN [76]. Accordingly, we derived three BL "outputs" which differ in their observation function that predicts the mismatch response.

- **1. Novelty detection or Surprise ($BL^{surp}$).** A first way to describe the MMN response in the Bayesian leaning framework is as a neuronal process encoding surprise about the category of the stimulus. It corresponds to the novelty detection model about sensory inputs described by Lieder et al. [76] and can be formulated as the Shannon surprise on the current input category: $-\ln p(u_i|x_i)$, with $p(u_i|x_i)$ being the probability of observing input $u$ at trial $i$ given prior belief on stimuli probability $x_i$.

$$g(x_i, u_i, \lambda) = \lambda_0 - \ln p(u_i|x_i)\lambda_1$$

$$\text{with } p(u_i|x_i) = \begin{cases} \dfrac{x_{1,i}}{x_{1,i} + x_{2,i}} & \text{if } u_i = 1 \ (\text{deviant}) \\[3mm] \dfrac{x_{2,i}}{x_{1,i} + x_{2,i}} & \text{if } u_i = 0 \ (\text{standard}) \end{cases} \tag{8}$$

- **2. Precision-Weigthed Prediction Error ($BL^{pwpe}$).** In this view, the brain would implement approximated Bayesian inference by doing predictive coding: hierarchical minimization of precision-weighted prediction errors [42]. In our modelling framework we could simulate this process by computing the difference between the prediction about incoming sensory data (i.e. the expected value of the prior Beta distribution) and sensory input $u_i$, weighted by the precision of the model (i.e. the inverse variance of the prior Beta distribution).

$$g(x_i, u_i, \lambda) = \lambda_0 + \frac{u_i - \mathrm{E}[B(x_{1,i}, x_{2,i})]}{\mathrm{var}[B(x_{1,i}, x_{2,i})]}\lambda_1 \tag{9}$$

- **3. Model adjustment ($BL^{madj}$).** Finally Bayesian learning can be conceptualized as the amount of belief updating about model parameters. It was described as "Bayesian surprise" in Ostwald et al. [71] and as "model adjustment" in Lieder et al. [76]. We adopted this last formulation and quantified updating as the divergence between prior and posterior beliefs about parameter $\mu$. In practice it can be computed as the Kullback-Leibler divergence $D_{KL}$ (or relative entropy) between posterior and prior Beta distributions over $\mu$, leading to the following observation function:

$$g(x_i, u_i, \lambda) = \lambda_0 + D_{KL}(\mu_{prior}, \mu_{posterior})\lambda_1$$

$$\text{with } D_{KL} = \int p(\mu_{prior}) \ln\left(\frac{p(\mu_{prior})}{p(\mu_{posterior})}\right) d\mu = \int p(\mu|x_{1,i}, x_{2,i}) \cdot \ln\left(\frac{p(\mu|x_{1,i}, x_{2,i})}{p(\mu|x_{1,i+1}, x_{2,i+1})}\right) d\mu \tag{10}$$

For Beta distributions $D_{KL}$ is simplified and can be solved analytically:

$$\begin{aligned} D_{KL} = \ & \log\left(\frac{\Gamma(x_{1,i} + x_{2,i})}{\Gamma(x_{1,i+1} + x_{2,i+1})}\right) + \log\left(\frac{\Gamma(x_{1,i+1})}{\Gamma(x_{1,i})}\right) + \log\left(\frac{\Gamma(x_{2,i+1})}{\Gamma(x_{2,i})}\right) + \\ & (x_{1,i} - x_{1,i+1})[\psi(x_{1,i}) - \psi(x_{1,i} + x_{2,i})] + (x_{2,i} - x_{2,i+1})[\psi(x_{2,i}) - \psi(x_{1,i} + x_{2,i})] \end{aligned} \tag{11}$$

with $\Gamma$ and $\psi$ the Gamma Euler and digamma Euler functions.

**Model fitting.** We ran VBA model inversion scheme on the 27 models defined before (null, DD, SSA with seven different $\tau_a$ and three BL outputs with six different $\tau_t$ each), independently for each subject and time window of epoched EEG data. We defined 21 non overlapping 10ms time windows from 45 to 255 ms around tone onset (45 to 55ms, 55 to 65ms, etc). We restricted the modeled interval to 50–250ms so as to alleviate the computational burden while covering the time window of statistical significance of the MMN usually found in the literature (100–200ms). The data supplied to the VBA model inversion function was the EEG activity averaged temporally over each 10ms window, and spatially over the ROI defined before, trial by trial.

Priors over models' parameters and initial hidden states need to be set before model inversion and were identical for both studies. All priors and initial hidden states were defined as multivariate normal distributions. No evolution parameters $\theta$ were used as for the SSA and BL model families, $\tau_a$ and $\tau_t$ were fixed for each different model and not fitted.

All models had two observation parameters $\phi$ (except the null model which only had one): a factor $\phi_1$ scaling the model response and an offset $\phi_0$. We set them empirically in the same manner for all models with prior means defined per subject by:

- $\phi_0 = mean(standards)$: the mean amplitude over all standard tones (in the given time window)

- $\phi_1 = mean(deviants) - mean(standards)$: averaged deviant minus standard amplitude, hence a measure of the MMN response (again in the time window of interest)

and a diagonal prior covariance matrix with variance 10.

The prior for the measurement noise precision $\sigma$ was a Gamma distribution of shape 0.1 and rate 1, consistent with a standard deviation (inverse precision) for amplitude measurement of 10 μV.

Initial hidden states ($X0$) are necessary to initiate the inversion and depended on the models. The null model did not have any hidden state. For all other models, a null covariance matrix was specified, encoding absolute certainty about the initial values of hidden states. The DD model's single hidden state was initiated to 0, while SSA models' initial hidden states were set to 1, reflecting the likely assumption that both standard and deviant specific neuronal populations are not adapted before hearing any tone and their responsiveness is therefore maximal. BL models' initial hidden states were also conventionally set to 1, even if no tones were ever heard, to satisfy computational constraint (the Kullback-Leibler divergence is not defined for null hidden states).

In study 2, all parameters were reinitialized and fitted independently for the two sessions, as these were acquired several hours apart. In Study 1, and within each session of Study 2, experimental conditions and blocks were separated only by short transitions of a few seconds (these were not considered in the modeling scheme). Therefore, the inputs from all conditions and blocks were used and treated as a continuous and uninterrupted sequence for the evolution function, but only the EEG data from the control conditions were used to inform model inversion. Similarly trials marked for rejection during pre-processing, as well as outliers defined as any trial whose EEG amplitude exceeded four times the standard deviation over all trials, were included in the sequence of inputs to modeled—informing the evolution function dynamics—but not in the computation of goodness-of-fit measures.

VBA inversion outputs the summary statistics (mean and covariance) of the posterior distributions of all model variables, as well as goodness-of-fit metrics of which two will be of particular interest to us:

- the model's variational free energy *F*—an approximation of model log-evidence that favors accuracy while penalizing model complexity [87]. It was shown to outperform other information-theoretic criteria for model comparison [88] and so will be used in this sense (see below).

- the percentage of variance explained by the model $R^2_{obs}$, which will allow us to derive an absolute goodness-of-fit measure.

**Inference.**   We carried out inference in two hierarchical steps. First we compared models at the level of families, i.e. sets of models sharing a similar structure or computational feature. In the present study the families are SSA and the three outputs of BL, and each of them aggregates models that differ only in the value of the temporal parameter $\tau$ ($\tau_a$ for SSA and $\tau_t$ for BL). This first step allowed us to select the most plausible computational mechanism for each (10ms averaged) post-stimulus time windows of the auditory evoked response. Second, we estimated the value of $\tau$ for the winning family through Bayesian model averaging (BMA). The two steps are described below in more detail.

**Bayesian Model Comparison (BMC).** There are two different approaches to BMC, depending on the assumption made about the structure of the underlying population. In fixed-effect BMC (ffx–BMC), the population is assumed to be *homogeneous* in the sense that all subjects' data are best described by the same unique model. Alternatively, random-effect BMC (rfx–BMC) treats the optimal generative model as a random effect that may vary across subjects, with a fixed but unknown *heterogeneous* population distribution.

Stephan et al. (2010) recommends to select the population structure (and, therefore, the BMC procedure) based on characteristics of the effect of interest, with the homogeneous assumption being "warranted when studying a basic physiological mechanism that is unlikely to vary across the subjects" [89]. The mechanisms underlying the MMN arguably falls under this definition, justifying the use of ffx-BMC. Alternatively, the choice between inference procedures may be data-driven as described in S1 Text.

In fixed-effect BMC group-level inference starts out by calculating the evidence for each model given the whole sample using the product rule for independent events:

$$p(y|m) = p(y^{(1)}, y^{(2)}, \cdots, y^{(N)}|m) = \prod_{s=1}^{N} p(y^{(s)}|m) \tag{12}$$

where $y^{(s)}$ is the observed data for the participant *s* and *N* is the number of participants. Group model evidence is often calculated in log form:

$$log(p(y|m)) = \sum_{s=1}^{N} \log p(y^{(s)}|m) \tag{13}$$

Then, the *posterior model probability* (PMP) of model $m_j$ can be derived from all models evidence using the Bayes theorem:

$$p(m_j|y) = \frac{p(y|m_j)p(m_j)}{\sum_{k} p(y|m_k)p(m_k)} \tag{14}$$

In the context of a uniform distribution over model space (i.e. the prior belief that all models are equally likely), which we will assume in the present study, the previous formula can be

further simplified to:

$$p(m_j|y) = \frac{p(y|m_j)}{\sum_k p(y|m_k)} \tag{15}$$

The model with the highest posterior model probability may then be selected as the most plausible at the population level. Alternatively, one may focus on the Bayes Factor for a model $j$ (compared to all other models) which can be written as the ratio of posterior odds to prior odds:

$$BF_{m_j,all} = \frac{p(m_j|y)}{1 - p(m_j|y)} \Big/ \frac{p(m_j)}{1 - p(m_j)} = \frac{p(m_j|y)}{1 - p(m_j|y)}(K-1) \tag{16}$$

with $K$ the number of compared models, all with equal prior probability $p(m_k) = 1/K$.

**Bayesian Model Averaging (BMA).** Once a model family has been selected, the best value for the $\tau$ parameter can be found by using a BMC procedure again, this time restricted to a single family. Alternatively it is possible to take into account the model uncertainty in the estimation of $\tau$. BMA does so by averaging all possible (modeled) values of the parameter, weighted by the posterior probability of the corresponding model:

$$\hat{\tau} = \sum_{\substack{m \in M \\ \tau \in T}} \frac{\tau \cdot p(m_\tau|y)}{|T|} \tag{17}$$

where $M$ designates a family of model, $T$ the set of $\tau$ values that were modeled, and $|T|$ the size of $T$.

**Interpretation of Bayes Factors.** We will adopt the conventional heuristics proposed by Kass and Raftery (1995) [90] whereby a BF higher than 100 is interpreted as decisive evidence, a BF between 10 and 100 indicates strong evidence and a BF between 3 and 10 suggests substantial evidence. BF below 3 are deemed inconclusive.

**Model recovery.**   Before taking the results of model comparison at face value, it is essential to validate that the BMC procedure delivers reliable results using simulations [91]. Model recovery analysis ascertains whether and under which experimental conditions (design, sample size, signal-to-noise ratio) we can conclusively arbitrate between different models. The overall approach involves generating synthetic data from each of the candidate models and fitting all the models to determine whether we can *recover* the true one. The main purpose of the model recovery analysis for the present study is to assess the separability of SSA from BL models, and the possibility to disambiguate BL outputs. Therefore we restricted the simulated model space to one specific value of $\tau_a$ (for SSA models) and one for $\tau_t$ (for BL models), making the analysis more tractable. Based on previous modeling results obtained from an oddball experiment by some of us [72], we chose $\tau_t = 20$. The three BL output functions are highly correlated at this value of $\tau_t$ (0.93 on average), more so than at most other parameter values, ensuring that the results of model recovery will be conservative. Following the same logic we chose the value of $\tau_a$ that maximized the correlation between SSA and $BL_{20}$ models, that is 10 (average $SSA_{10}$–$BL_{20}$ correlation = 0.94).

The details of our model recovery analysis are depicted in Fig 1. For each participant, the input sequence that was presented to her is fed to a candidate model to generate a series of predicted responses (Fig 1A), which is then declined in multiple versions varying in amplitude of the predicted deviant minus standard difference response, as well as in level of Gaussian noise (Fig 1B). Amplitude and noise levels were manipulated factorially, resulting in 30 pairs chosen to span the range of SNR values observed in the electrophysiological data during the MMN

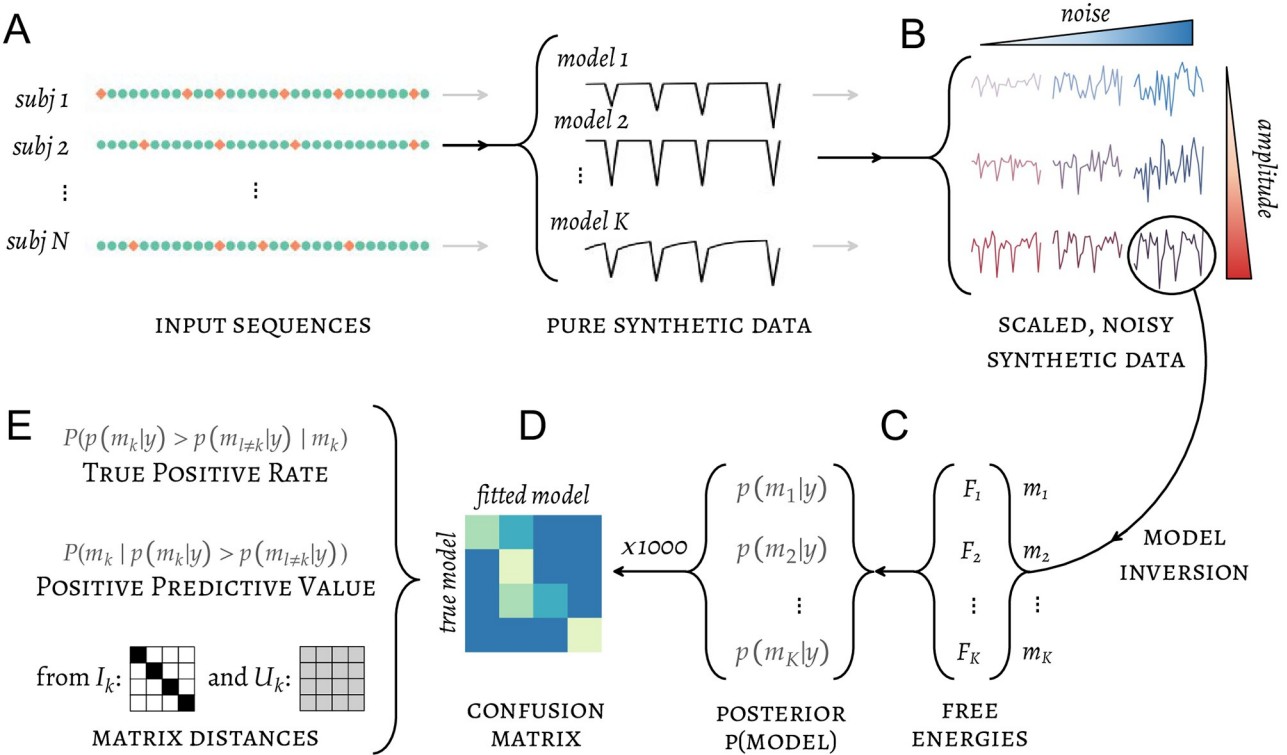

**Fig 1. Model recovery analysis pipeline.** (A) The sequences of input presented to participants are fed to the candidate models to generate sequences of synthetic data which are subsequently (B) scaled and noised in multiple ways to cover a range of signal amplitudes and intertrial variances. The large number of synthetic datasets thus obtained can be used to constitute unique samples with an arbitrary distribution of amplitude and variance. (C) Each such sample is subjected to model fitting (free energies) and group-level inference (posterior model probabilities). (D) The results of model selection from 1000 simulated samples are reflected in a single confusion matrix, which may be (E) summarized in quantities of interest such as the distance from ideal matrices or indices of sensitivity and specificity.

time window. Besides, 5 datasets were generated for each (amplitude, noise) pair—these datasets shared the same statistics and differed only in the actual realization of noise. Therefore, a total of $150 \times K \times N$ datasets were generated, where $K$ denotes the number of candidate models and $N$ the number of participants. The candidate models were then fitted on these synthetic datasets, yielding free-energy approximations of model evidence that were used to select the best model as described before, i.e. the model with the highest posterior probability (Fig 1C). In practice we carried out BMC on 1000 bootstrapped samples. Each of these samples was composed of $N$ participants' data simulated at the appropriate levels of amplitude and noise, each randomly draw from one of the 5 realizations of noise—allowing for $5^N$ possible combinations. The results from the 1000 simulations can be conveniently summarized and displayed in a confusion matrix $\mathbf{A} = (a_{ij})$ where the element $a_{ij}$ denotes the frequency with which each candidate model $j$ has been selected for data simulated under model $i$ (Fig 1D). The confusion matrix is of size $K \times K$ but may be summarized in fewer pieces of information (Fig 1E). Of particular interest are two performance indices that are defined model-wise: the *true positive rate* (TPR) which is the chance we have to select the true model as the best one assuming that it is included in the model space, and the *positive predictive value* (PPV) which is the chance that a model selected as the best is indeed the true model (again, assuming that the latter is in the model space). TPR and PPV are measures of sensitivity and specificity, respectively, with TPR amounting to statistical power in a 2-class problem. Ideally if the BMC procedure worked

perfectly the confusion matrix would be the $K \times K$ identity matrix and we would have TPR = PPV = 1. Thus, another useful summary statistic of a confusion matrix is its distance from the optimal, identity matrix $\mathbf{I_K}$—which we computed using the $L_1$ norm:

$$|\mathbf{A} - I_K|_1 = \sum_{i=1}^{K}\sum_{j=1}^{K}|a_{ij} - \delta_{ij}| \text{ with } \delta_{ij} = \begin{cases} 1 \text{ if } i = j \\ 0 \text{ otherwise} \end{cases} \tag{18}$$

Conversely, if models were not distinguishable and the model selection produced random results under the experimental design and other relevant characteristics of the data, the confusion matrix would be a matrix $\mathbf{U_K}$ with all elements equal to $1/K$ (hereafter called the *normalized unit matrix*) and we would have TPR = PPV = $1/K$. The randomness of the confusion matrix may be summarized by its distance from the normalized unit matrix.

Confusion matrices, as well as their distance to identity and normalized unit matrices are displayed as a function of SNR (magnitude/noise pairs) and interpreted in the "Model recovery" section of the results. In order to get a hint of our ability to decide between different models in our real data along the timecourse of the MMN response, we selected confusion matrices matching the level of SNR in EEG data for each participant, at each post-stimulus time window (50–250ms). Time-resolved TPR and PPV metrics were then computed based on "real SNR" and compared to conventional thresholds.

**Absolute goodness-of-fit.** BMC procedures pick a model out of a set of candidates based only on their *relative* performance, i.e. their comparative ability to predict the observed data while being parsimonious. Said otherwise, model comparison can single out a model as being better at predicting the data than other examined models, but it does not follow that the selected model is good in an *absolute* sense. It might actually be missing important features of the data, suggesting that the true generative model (or a useful approximation of it) was not in the set of candidate models and remains to be discovered.

A simple and straightforward measure of *absolute* fit is the coefficient of determination $R^2$, which in the context of linear regression, can be interpreted as the proportion of variance in the data accounted for by the fitted model. However, there is no universal threshold for $R^2$ that qualifies a model as "sufficiently good". In the context of the phenomenon of interest here, the variance in EEG-recorded neural activity can be divided in two sources: variance due to history-dependent processing of the controlled auditory stimuli, and variance due to uncontrolled factors (background neural and physiological activity and measurement noise). Only the former is the target of the modelling efforts and can theoretically be accounted for at 100% by a perfect model—but its relative importance with respect to the total variance remains unknown.

In order to estimate the proportion of total variance in the EEG signal that could be attributed to stimulus and regularity processing, we used the models to generate pure synthetic datasets whose variance, by construction, can be entirely attributed to stimulus processing. This variance could then be compared to the total variance observed in the data at a given latency $t$, yielding an ideal coefficient of determination $R_{max}^2$, i.e. the maximum proportion of variance that can be expected to be accounted for by a model $m$, assuming that this model has generated the data. $R_{max}^2$ can be expressed as a ratio of sum-of-squares:

$$R_{max}^2(m, t) = \frac{SS_{model}}{SS_{total}} = \frac{\sum_i (g_{m,i} - \bar{g}_m)^2}{\sum_i (y_{t,i} - \bar{y}_t)^2} \tag{19}$$

where $g_m$ is the observation function of model $m$ (see Eq 3), $y_t$ the EEG data at latency $t$, $i$ indexes the trials and the overline designates the mean across all trials. The actual observed

coefficient of determination $R^2_{obs}(m, t)$ can then be compared to this ideal index:

$$R^2_{obs}(m, t) = 1 - \frac{SS_{res}}{SS_{total}} = 1 - \frac{\sum_i \epsilon^2_{m,i}}{\sum_i (y_{t,i} - \bar{y}_t)^2} \tag{20}$$

with $\epsilon_{m,i}$ the Gaussian measurement noise for model $m$ at trial $i$ (see Eq 3).

Note that $R^2_{obs}$ may be greater than $R^2_{max}$ by chance alone, e.g. if the actual realization of noise positively correlates with the predictions of the generative model. Therefore, a conclusion may be reached only by examining the distribution of $R^2_{obs}/R^2_{max}$ at the group level.

## Results

In order to test whether different mechanisms underlie the MMN at various latencies of the difference potential, we fitted candidate computational models on EEG data collected during passive oddball experiments. Each of these models was designed to implement a prevailing account of the MMN and generates predictions of the trial-to-trial dynamics of the auditory ERP. Models were fitted independently on 10ms–wide averaged post-stimulus time windows ranging from 45 to 255ms using a Bayesian modeling framework. Thereafter we will refer indistinctly to these temporally averaged windows as "(post-stimulus) time windows" or "latencies". The free energy approximation to model evidence and the coefficient of determination $R^2$ were used as measures of relative and absolute goodness-of-fit, respectively.

In the first section we validated the relevance of the tested models by comparing them to a null model devoid of any dynamics. We then analyzed the relationship between model evidence and SNR to explore the relationship between data quality and inferential value.

In the second section, and before moving on to the actual comparison of models, we performed a simulation analysis to assess the identifiability of models under the two studies designs and the effect of SNR on the performance of model recovery.

In the third section we compared the candidate models between them in two steps, starting with fixed-effect BMC at the level of families, with each family corresponding to a theoretical account of the MMN. Models within each family share the same computational rules and are differentiated only by a parameter $\tau$ governing the temporal scale of the dynamics, which was estimated in a second step through Bayesian model averaging.

Finally, we used an absolute measure of goodness-of-fit to evaluate how accurately the previously selected models can generate the EEG data.

### Model evidence

As a first step we validated the ability of implemented models to capture the general features of trial-to-trial variations in the EEG signal better than simple control models. We used group Bayes Factors (BF) between the models of interest on one hand, and the null model on the other hand, as a measure of model relevance. In both studies and for almost all models, BF values exceeded levels corresponding to decisive evidence against the null in time windows showing a significant MMN, and dropped down rapidly outside that range (Fig 2).

The influence of the temporal parameter $\tau$ varied across model families. For the SSA and $BL^{madj}$ families, the BF of models with fast dynamics (low $\tau$) tended to rise, peak or decrease later than models with higher $\tau$. As far as the maximum BF is concerned, intermediate values of $\tau_t$ (20–30) obtained slightly higher model evidence than others within the $BL^{madj}$ family, but no consistent pattern could be found across the SSA family for $\tau_a$. Finally, the value of $\tau_t$ appears to have little to no influence on model evidence for $BL^{pwpe}$ and $BL^{surp}$ models.

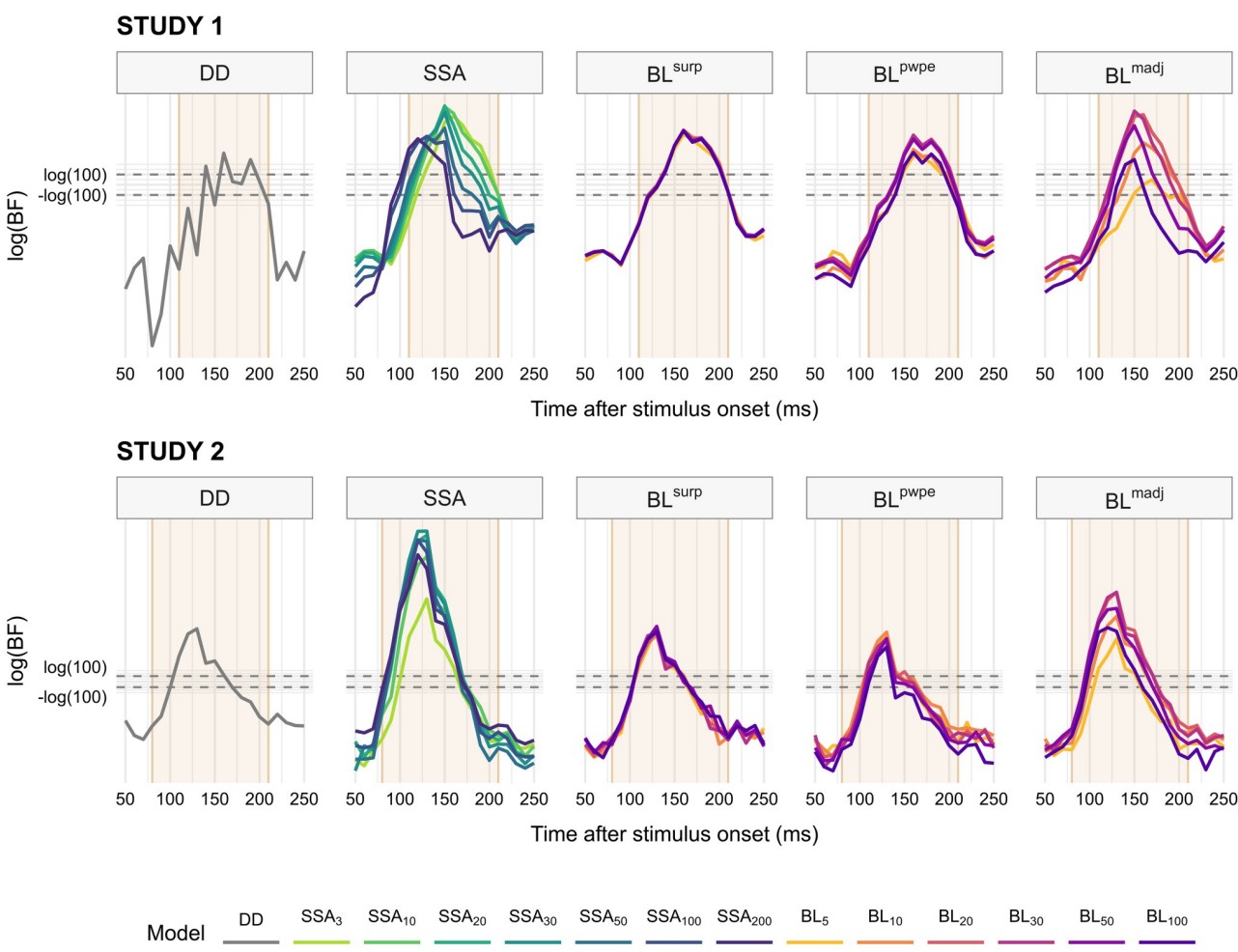

**Fig 2. Group-level Bayes Factors of models of interest against the null model.** Positive and negative values indicate evidence against and in favor of the null model, respectively. Horizontal lines represent Jeffreys' conventional thresholds for decisive evidence (BF = 100). For each study, the time range for which the deviant – standard difference potential, i.e. the MMN, is statistically significant is highlighted. Based on the definition of the model evidence, the group Bayes Factor (BF) is the product of individual BF, therefore the group log(BF) is the *sum* of individual log(BF). *DD*: deviant detection, *SSA*: stimulus-specific adaptation, *BL*: Bayesian learning, *surp*: Shannon's surprise, *pwpe*: precision-weighted prediction error, *madj*: model adjustment.

**Sensitivity to signal-to-noise ratio.** Fig 2 represents the Bayes Factor of models at the level of the whole sample, which is obtained by aggregating Bayes Factors of individual participants. However, the examination of individual model log-evidence reveals a widespread, strongly skewed distribution with many participants' values clustering around 0 or even negative (i.e. in favor of the null) and only a minority of the participants displaying substantial evidence against the null (e.g. about a third of the sample for SSA models; Fig 3 upper plots). While this pattern could result from inter-individual variability in MMN mechanisms, e.g. in prevailing time constants, such high individual selectivity is rather unlikely. We explored the alternative and more parsimonious hypothesis that the quality of the recorded data—indexed by the signal-to-noise ratio (SNR)—could partly condition the evidence against the null at the individual level.

The lower plots of Fig 3 represent a linear model of the relationship between log(BF) and SNR for a couple of selected models. There appears to be a strong association between the two

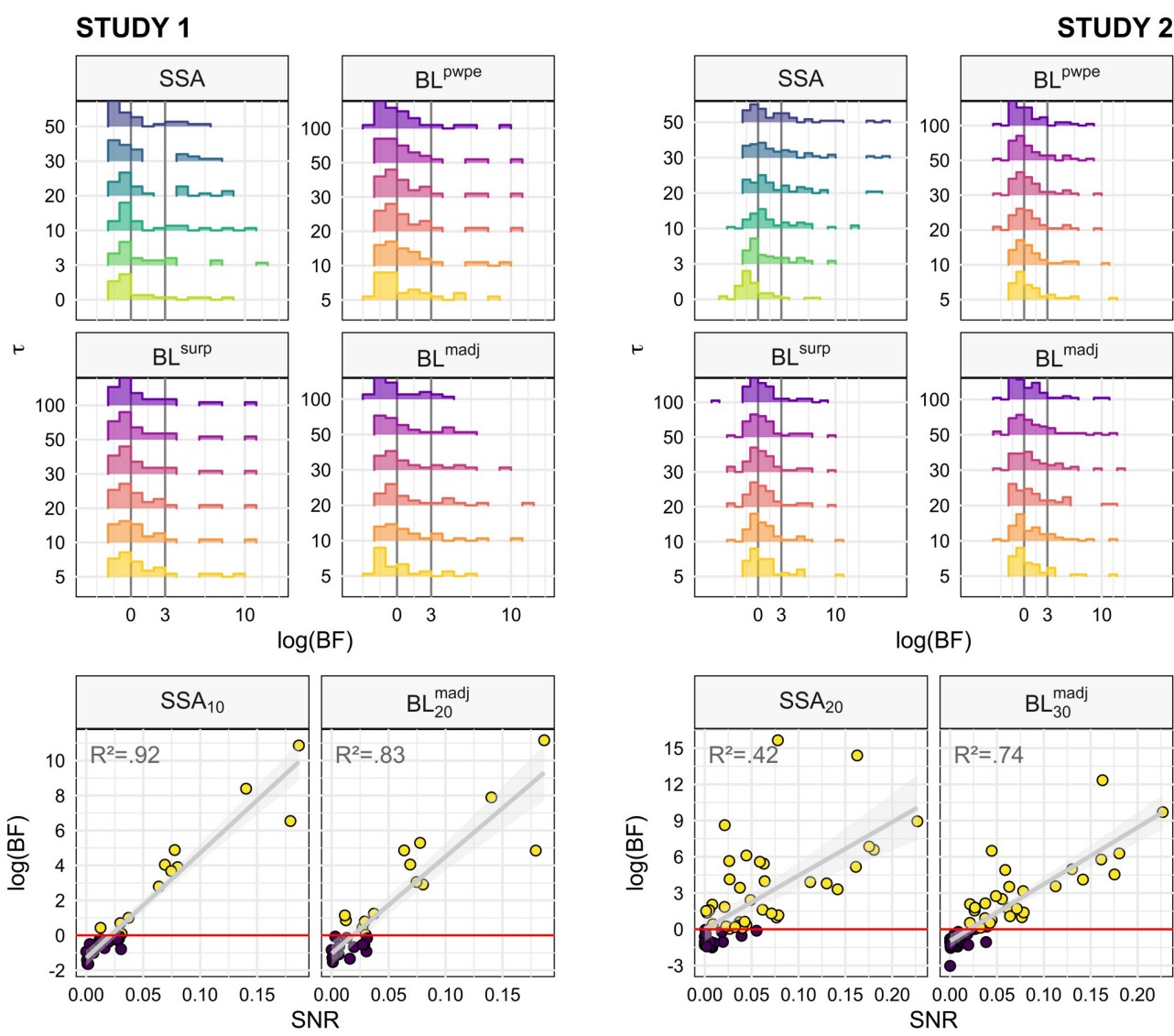

**Fig 3. Bayes Factors distributions and sensitivity to signal-to-noise ratio.** For each study, the upper plots depict the distribution of participants' model evidence for each value of $\tau$ within each family of models at the MMN peak latency. Vertical lines indicate thresholds for substantial and strong evidence, respectively. The lower plots represent the relationship between log(BF) and SNR for a couple of representative models. The values of $R^2$ refer to the fraction of variance modeled by a linear function. *DD*: deviant detection, *SSA*: stimulus-specific adaptation, *BL*: Bayesian learning, *surp*: Shannon's surprise, *pwpe*: precision-weighted prediction error, *madj*: model adjustment.

variables, with a simple linear function of SNR predicting a large portion of log(BF) variance, as indicated by the high $R^2$ values. The value of SNR above which log(BF) is predicted to be positive is around 0.02–0.03.

## Model recovery

**Sensitivity to signal-to-noise ratio.** Before performing model comparison and selection we assessed the ability of such procedure to disambiguate the candidate models given our experimental designs and levels of signal (MMN amplitude) and noise (spontaneous trial-to-trial variability). We simulated data based on stimuli sequences provided to the actual

participants and for a wide range of signal statistics values, and performed BMC on the synthetic data to evaluate model recovery and confusion. The results are displayed in Fig 4. Models can be perfectly recovered at the highest levels of SNR (Fig 4A), despite models being strongly correlated (see S4 Fig). However, identification gets poorer when noise increases or signal power decreases. The first models to be affected by degraded signal quality are $BL^{surp}$ and $BL^{pwpe}$ which get mixed up, consistent with them being the most correlated models (r > .99). With even poorer SNR, model selection becomes biased rather than random, as indicated by the fact that the confusion matrix gets farther away from the optimal, identity matrix while remaining quite distinct from the random-selection, uniform matrix (Fig 4B and 4C). At the lowest SNR, $BL^{surp}$ tends to come out as the winning model independently of the true generative mechanism, especially in Study 2 (in Study 1, dominance is shared between $BL^{surp}$ and $BL^{madj}$).

**Sensitivity to sample size.**   The results above were obtained with the same number of participants than found in the collected datasets. Fig 5A illustrates what happens when the sample size decreases for one representative combination of MMN amplitude and noise level. Unlike with SNR (Fig 5B), decreasing sample size blurs the confusion matrix, rendering model selection more and more random.

**Sample-based model recovery.**   Fig 6 summarizes the results of the model recovery analysis based on latency-dependent levels of noise and MMN amplitude that were observed in our samples. As expected the confusion matrices (top row) are the closest to the optimal, identity matrix at latencies matching the peak of the MMN, typically around 150ms. Consistently, the sensitivity (TPR) and specificity (PPV) indices peak at the same time for most models. At earlier and later latencies confusion matrices stray away from the optimal structure with noticeable differences between the two studies: in Study 1 the confusion matrix becomes blurrier with the consequence that all models lose both sensitivity and specificity, whereas in Study 2 the confusion matrix remains sharper but exhibits a strong bias in favor of $BL^{surp}$ and at the expense of SSA and $BL^{pwpe}$. This bias is present to some extent throughout the entire time range and leads to poor positive predictive value of the $BL^{surp}$ model in Study 2.

## Group-level inference

**Model comparisons at the family level.**   In the first section we established that all candidate generative models of the MMN are plausible under the collected EEG data. The question that then arises is whether some models are superior to others in their ability to explain the observed data. There are two approaches to Bayesian group inference depending on whether the targeted population is homogeneous—i.e. one may assume that all participants' data have been generated by the same model—or heterogeneous [89]. For such fundamental and automatic cognitive processes as low–level perceptual learning, it is reasonable to assume that their fundamental architecture and biological mechanisms have been primarily shaped by evolution, and as such are hard-coded rather than experience-dependent. Following this logic, one may presume that all participants in an oddball experiment share the same generative mechanism(s) for the MMN, justifying a fixed-effect approach to group-level in Bayesian modeling. Alternatively, one may sidestep such speculations and rather choose to approach the issue of population structure as an empirical question that may itself be addressed using the Bayesian inference framework. We implemented this data-driven approach and obtained results aligned with the theoretical assumption of homogeneity (see S1 Text and S5 Fig).

We first compared the four families of interest—SSA, $BL^{surp}$ $BL^{pwpe}$ and $BL^{madj}$—along with a control family that included null and DD models (Fig 7). We found that SSA models dominated the first part of the MMN time-window, from the moment it could be disambiguated

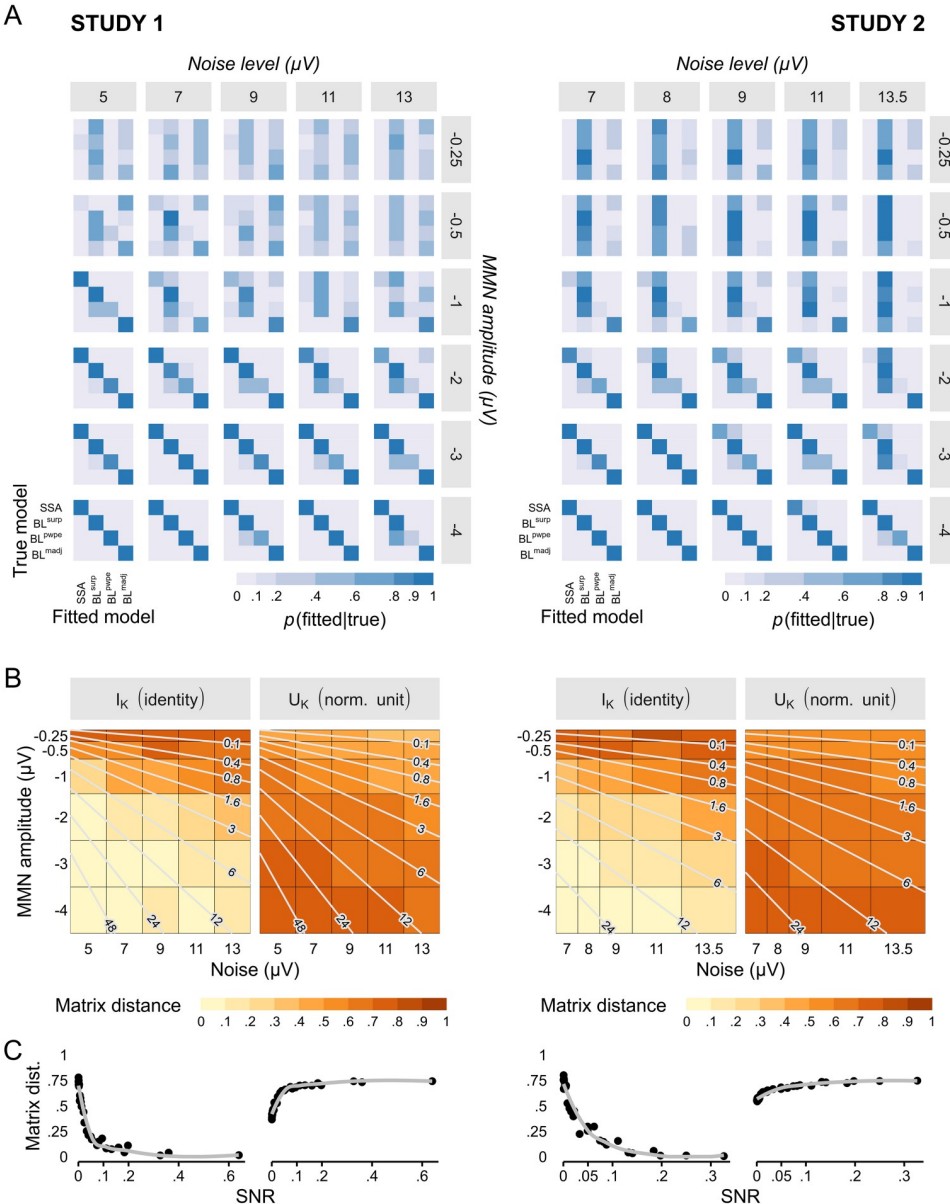

**Fig 4. Model recovery analysis in the noise-amplitude grid space.** The noise-amplitude grid was defined so as to cover the range of observed noise levels and signal amplitude, which was slightly different between the two studies. (A) The confusion matrix is perfect at the highest SNR levels (high amplitude and low noise, bottom left) but gets increasingly degraded at lower SNR (low amplitude or high noise, upper right corner of the grid) towards biased model selection. (C) The effect of poor SNR can be quantified using the matrix distance between the confusion matrix and the optimal, identity matrix (left) or the uniform matrix (right). Matrices that are far from the identity matrix are those with low diagonal elements, corresponding to poor recovery of the true model in favor of confounding competitors, but not necessarily random ones. Matrices that are far from the uniform matrix correspond to deterministic identification, i.e. the same model tends to win systematically under a given generative model, but not necessarily the correct one. Light lines indicate isocontours of SNR values. (C) Matrix distances as a function of SNR. At low SNR, the high distance from both the identity and the uniform matrix indicate that confusion matrices become biased rather than random.

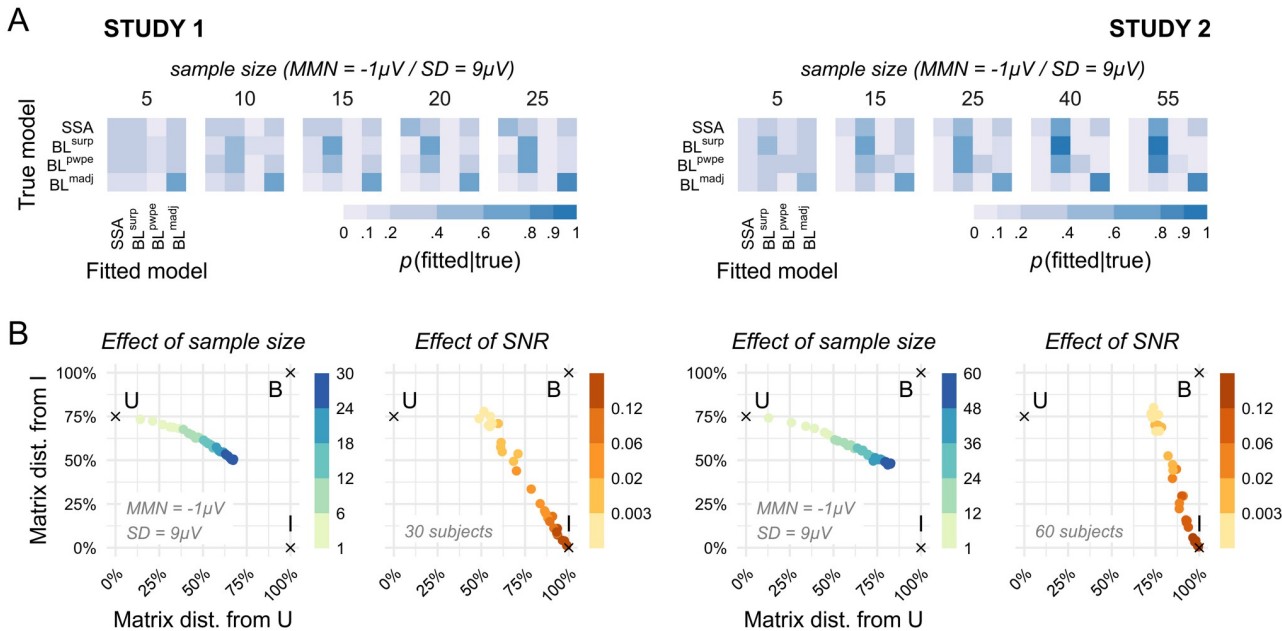

**Fig 5. Influence of sample size on model recovery.** (A) Decreasing sample size has the effect of degrading the confusion matrix towards more "blurred" (i.e. uniform) matrices. (B) Representing confusion matrices by their distance from the identity and the uniform matrices (I and U, respectively), emphasizes the specific influences of sample size and SNR on model discrimination. While low SNR leads to highly precise—i.e. deterministic—but biased confusion matrices (marked as B on the graph), low sample size results in random model selection.

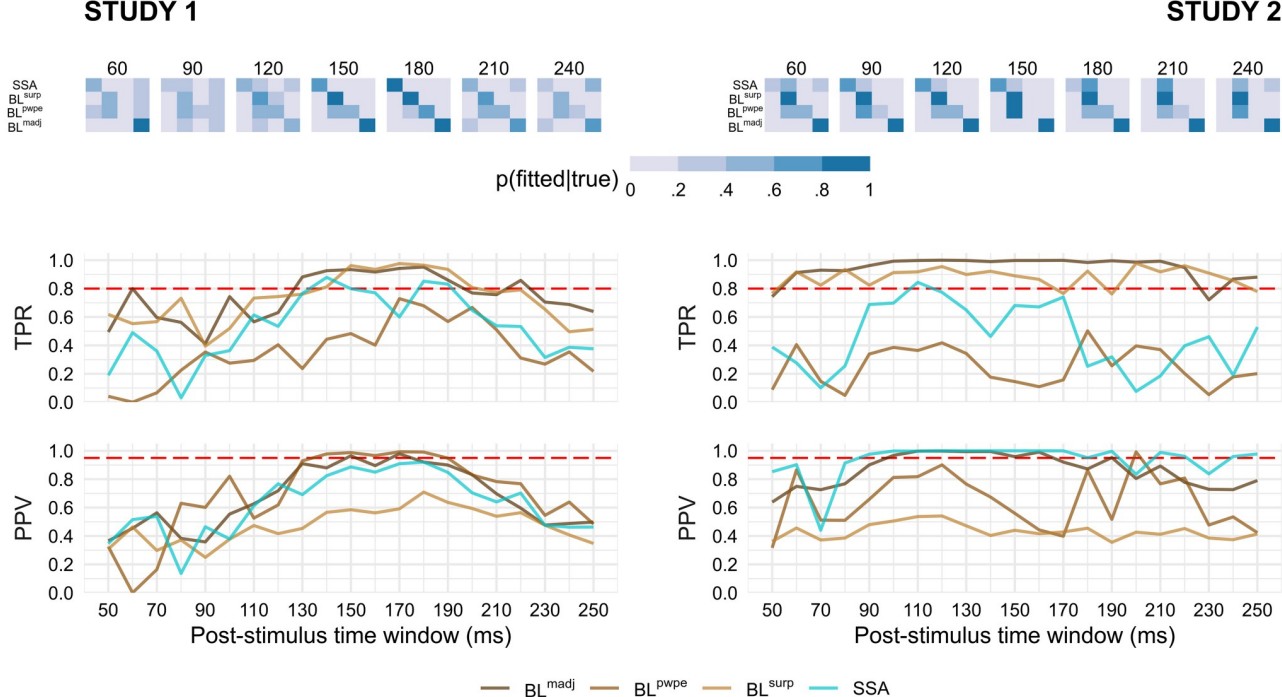

**Fig 6. Time-resolved model recovery analysis.** (Top) Confusion matrices were constructed for each latency using simulations matching the noise level and signal amplitude of each participant. The true positive rate (TPR, middle) and the positive predictive value (PPV, bottom) were then extracted for each model. Dashed red lines indicate conventional thresholds: 0.8 for TPR (corresponding to 80% statistical power) and 0.95 for PPV (analogous to a type I error rate of 5%).

from pure noise around 80–90ms until 150–160ms. SSA models then overlapped with $BL^{madj}$ for 20ms, before finally making way for $BL^{surp}$ and $BL^{pwpe}$ models, especially in Study 1. Importantly, the general temporal organisation of models was largely similar across the two independent studies, suggesting that this pattern of results is replicable and reliable. Discrepancies between the studies appeared only at the latest latencies and will be discussed in the next paragraph.

**Simulation-based assessment of reliability.** The results from the simulation-based, model recovery analysis are of great value to understand and assess the reliability of the patterns observed in the real data depicted in Fig 7. The first remarkable result is the domination of SSA models from the onset of the MMN until about 150ms. From the model recovery analysis SSA appears to have very high positive predictive value (>.95) on the entire MMN time window in Study 2, and acceptable PPV (>.80) between 140 and 180ms in Study 1 (Fig 6), suggesting that the dominance of SSA is most likely a true finding rather than a confusion from BL models. A similar reasoning may be applied to the moment when $BL^{madj}$ starts competing with SSA: at these latencies $BL^{madj}$ has a high PPV (>.90) according to the simulations, and the probability that it confounds SSA is low in both studies (Study 1: $p(BL^{madj}|SSA) < .10$ at t = 160–170ms; Study 2: $p(BL^{madj}|SSA) < .14$ at t = 170–180ms). Importantly, the reciprocal was true at these latencies: SSA had a low chance to be selected as the winning model under $BL^{madj}$ in the simulations ($p < .09$ in Study 1; $p < .02$ in Study 2). All in all the simulation analysis suggests that the very close posterior probabilities of model families SSA and $BL^{madj}$ around 170ms are unlikely to be driven by a lack of separability, but may rather reflect a temporal overlap of the two modeled mechanisms. Lastly, the two studies diverge with respect to the dominant family at the latest latencies with Study 1 favoring $BL^{surp}$ and Study 2 showing

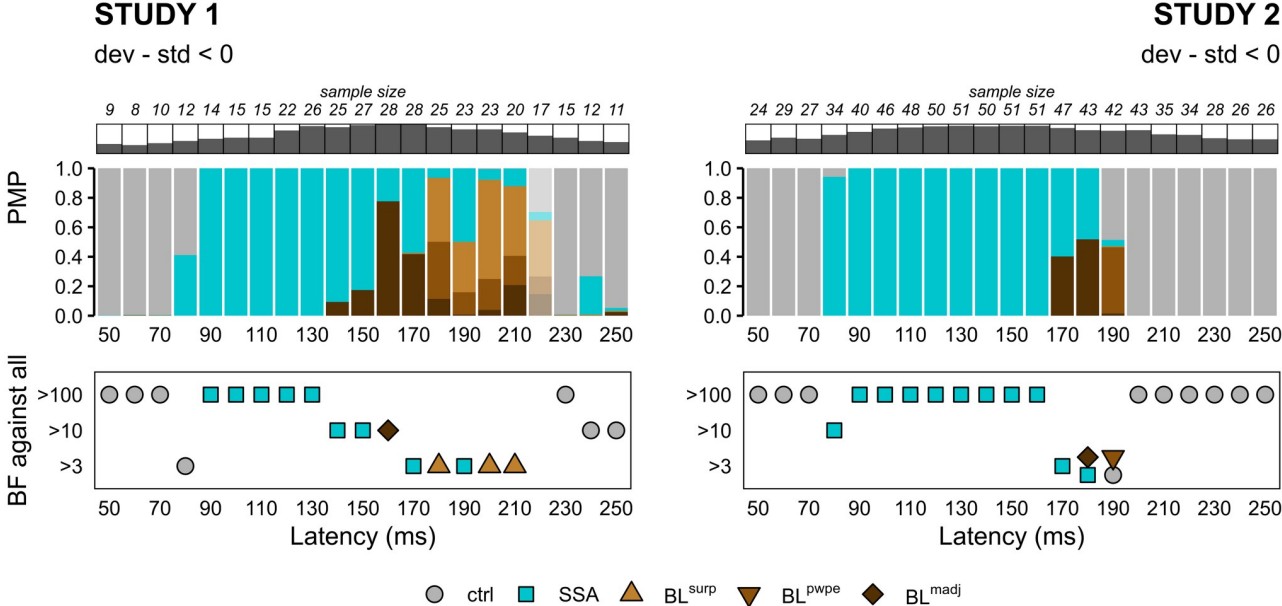

**Fig 7. Family-wise fixed-effect BMC.** (*Top*) Posterior model probabilities (PMP) at the family level for each study and each post-stimulus latency. Latency-specific sample sizes are indicated above PMP plots along with dark bars representing the fraction of the study sample retained. Shaded PMP bars correspond to latencies for which no model family dominate substantially (BF <3). (*Bottom*) Relative evidence of each model family against all other families. Only families with a substantial level of evidence (*BF* > 3) are displayed. In practice, most of the time there was one and only one family with substantial evidence at each latency, except at t = 180ms and t = 190ms in Study 2. *ctrl*: control models (null and deviant detection), *SSA*: stimulus-specific adaptation, *BL*: Bayesian learning, *surp*: Shannon's surprise, *pwpe*: precision-weighted prediction error, *madj*: model adjustment.

evidence for $BL^{pwpe}$. Unfortunately these two BL outputs are the most correlated (r >.99) and are not easily disambiguated, even under good SNR conditions.

**Model comparisons in high quality samples.** Our exploration of the effect of SNR on model identifiability (Fig 4) suggested that the presence of low SNR participants is far from innocuous: not only does it impact negatively statistical power—as one would expect from any source of noise that does not contribute useful information but is permitted to weigh on the inferential process—but it also biases the model selection procedure. An optimistic way of looking at it is that the sensitivity and reliability of the inference should improve when it is focused on those participants that exhibit the highest SNR, so long as the sample size remains sufficient. We explored this possibility by restricting the family-wise BMC analysis to latency-specific subsets of participants exceeding certain SNR thresholds. The results are depicted in Fig 8 and indicate that subsetting to high SNR participants has largely common but also some diverging effects across the two studies. In both studies adaptation (SSA) dynamics are detected earlier and BL dynamics up to 230ms post-stimulus. This improvement in time range shows even after a slight thresholding (SNR >0.001) in Study 1 but is more gradual in Study 2. The general temporal organization of models does not change significantly in either studies compared to the full sample, but while it is markedly reinforced in Study 2 with a clearer and extended dominance of $BL^{madj}$ followed by $BL^{surp}$ in the most restricted samples (SNR >0.008 and >0.024), it is partially attenuated in Study 1 with the elimination of $BL^{madj}$ at SNR >0.024. Here the discrepancies between the two studies might be due to differences in sample size, with the smaller number of participants in Study 1, especially after stringent selection, negatively impacting model identifiability to the point of canceling the gain offered by the increased sample SNR.

**Parameter estimation.** In the present study we have adopted a fixed parameter approach whereby the value of the $\tau$ parameter was fixed in submodels of the SSA and BL families rather than fitted. We used Bayesian model averaging (BMA) to estimate the value of the parameter by integrating over the family model space taking into account the uncertainty of the submodels as indexed by posterior model probability. The results for the adaptation parameter of SSA models and the integration constant of BL models are quite distinct (Fig 9). For the former, we found a consistent decrease of $\tau_a$ through time in both datasets—although less marked for Study 2 at higher SNR. This suggests an increasing gradient of adaptation (see relationship between $\tau_a$ and responsiveness in Eq 6) along the MMN timecourse which echoes electrophysiological results in rats and will be discussed later. On the other hand the time constant for BL models in the later stage of the MMN was rather constant through time.

## Absolute goodness-of-fit

The average proportion of total single-trial EEG variance accounted for by the best model across all subjects was similar across studies but depended on the MMN latency, ranging from 0.5 to 1.5% in Study 1, and from 0.6 to 1.8% in Study 2 (Fig 10, upper plots). While these figures may seem low at first glance they are consistent with previous results (1% of LFP data variance explained in [92]) and with the fact that, in EEG recordings background stimulus-unrelated activity and other sources of noise drown the event-related potentials by an order of magnitude or higher, requiring numerous presentations of the same stimulus and averaging of the resulting data to obtain an informative signal.

To gain an accurate sense of how adequate is the observed proportion of variance accounted for by a given model (denoted $R^2_{obs}$), we compared it to the expected value (denoted $R^2_{max}$) calculated from simulations that assumed the said model truly generated the data. $R^2_{max}$ is subject–, model– and latency– dependent. By definition $R^2_{obs}$ is expected to be lower to $R^2_{max}$ on

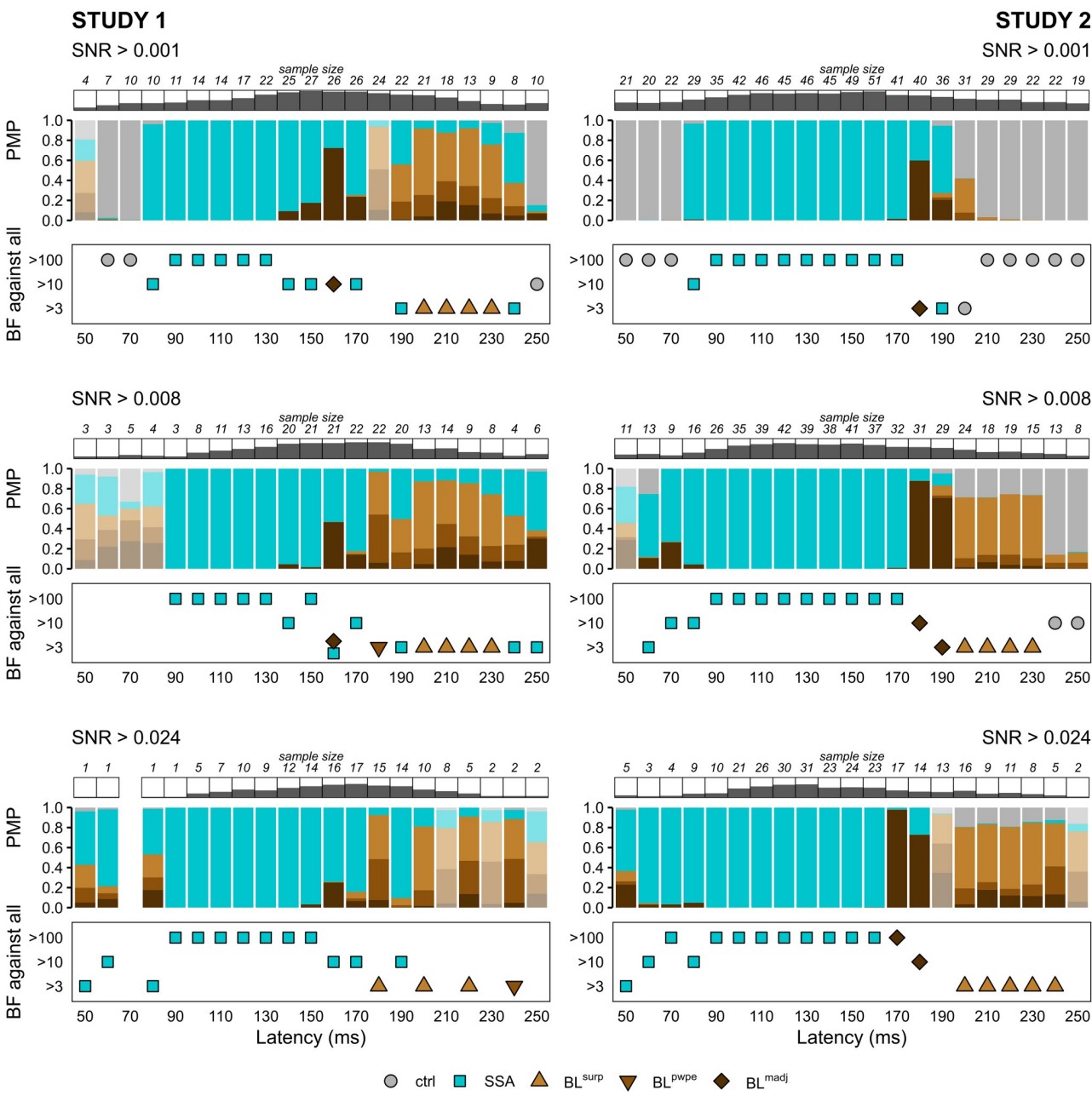

**Fig 8. Family-wise fixed-effect BMC in high quality samples.** Posterior model probabilities of model families (PMP) and relative evidence against all other families (BF) in subsets of samples defined by SNR exceeding 0.001 (top), 0.008 (middle) and 0.024 (bottom). Latency-specific sample sizes are indicated above PMP plots along with dark bars representing the fraction of the study sample retained. Shaded PMP bars correspond to latencies for which no model family dominate substantially (BF <3). *ctrl*: control models (null and deviant detection), *SSA*: stimulus-specific adaptation, *BL*: Bayesian learning, *surp*: Shannon's surprise, *pwpe*: precision-weighted prediction error, *madj*: model adjustment.

average. A low $R^2_{obs}/R^2_{max}$ ratio indicates that the model is suboptimal and is, at best, only an approximation of the generative process. Conversely, a ratio close to 1 suggests that the model can account for all the variance produced by the generative process and is indistinguishable from it under the circumstances of the study. This is indeed the case for the set of latency-dependent models that have been selected by the fixed-effect BMC procedure, as we found

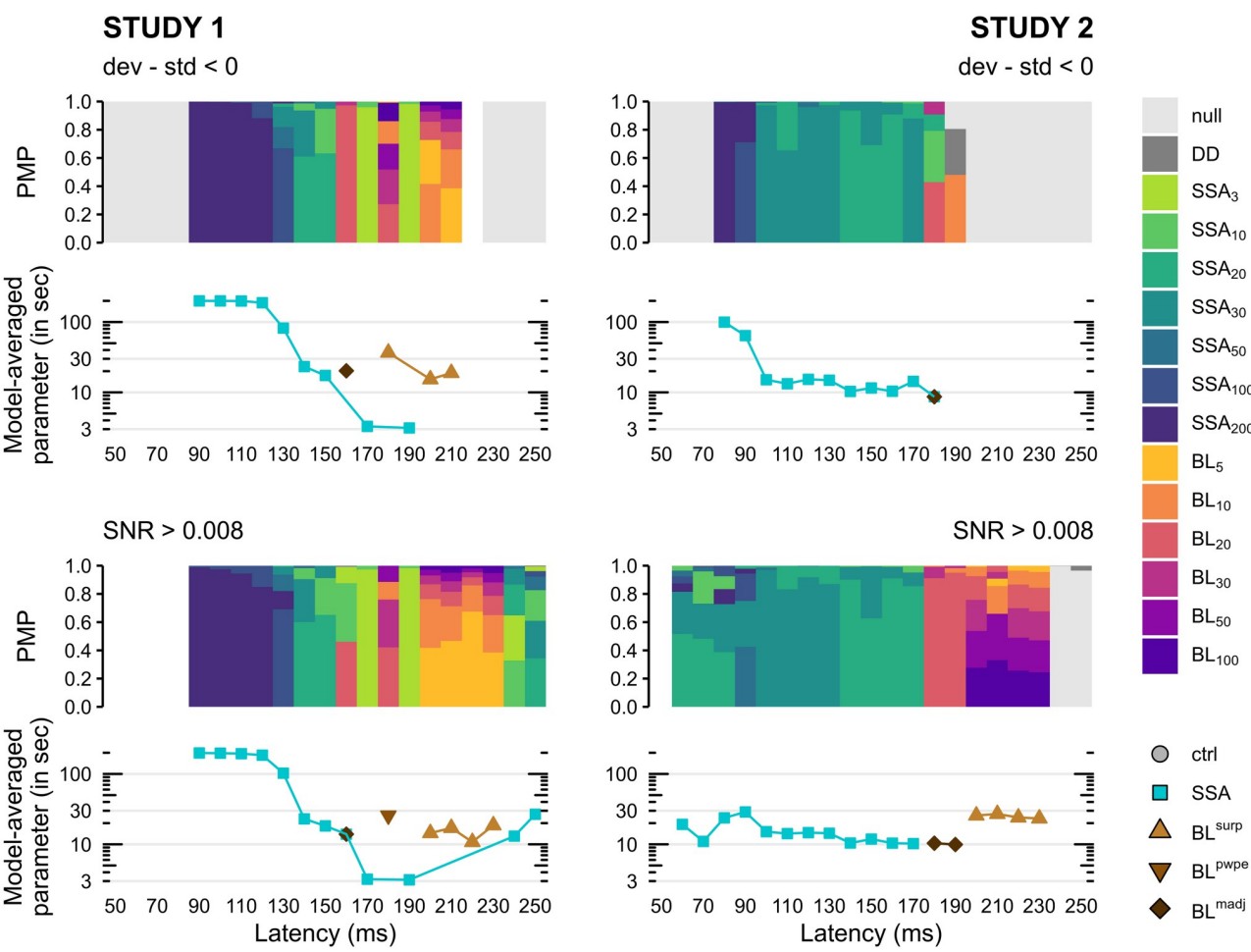

**Fig 9. Within-family Bayesian model averaging of the temporal parameter $\tau$.** (*Bar plots*) Posterior model probabilities (PMP) at the model level within the best family for each study and each post-stimulus latency without SNR filtering (top panel, dev-std <0, see Fig 7) and with only SNR >0.008 samples (bottom panel, see Fig 8). Colored bars are stacked from bottom to top in decreasing order of PMP. Values of $\tau$ are in stimulus units. (*Line graphs*) Estimated values for the time constant $\tau$ obtained by model averaging, i.e. averaged over all models of a family weighted by their respective PMP. Values are in seconds. *ctrl*: control models (null and deviant detection), *SSA*: stimulus-specific adaptation, *BL*: Bayesian learning, *surp*: Shannon's surprise, *pwpe*: precision-weighted prediction error, *madj*: model adjustment.

that $R^2_{obs}/R^2_{max}$ was not significantly different from 1 in either studies (Study 1, between 90 and 220ms: estimate = 1.03, 95% CI [0.95, 1.12], t(103) = 0.81, p = .42; Study 2, between 90 and 180ms: estimate = 1.03, 95% CI [0.93, 1.14], t(136) = 0.51, p = .61).

As a sanity check and validation of the $R^2_{obs}/R^2_{max}$ ratio, we calculated it for some models at latencies for which they were *not* the best. In Study 1, models that dominated the earliest and latest latencies ($SSA_{50}$ and $BL^{surp}_5$, respectively) had a significant low ratio at other latencies ($SSA_{50}$, between 130 and 220ms: estimate = 0.66, 95% CI [0.57, 0.77], t(79) = -5.46, p<.001; $BL^{surp}_5$, between 90 and 190ms: estimate = 0.92, 95% CI [0.88, 0.97], t(86) = -3.46, p<.001). The same pattern was found in Study 2 for models that dominated the early and late MMN latencies ($SSA_{50}$, between 100 and 190ms: estimate = 0.78, 95% CI [0.70, 0.88], t(131) = -4.36, p<.001; $BL^{madj}_{30}$, between 90 and 160ms: estimate = 0.84, 95% CI [0.76, 0.92], t(119) = -3.88, p<.001).

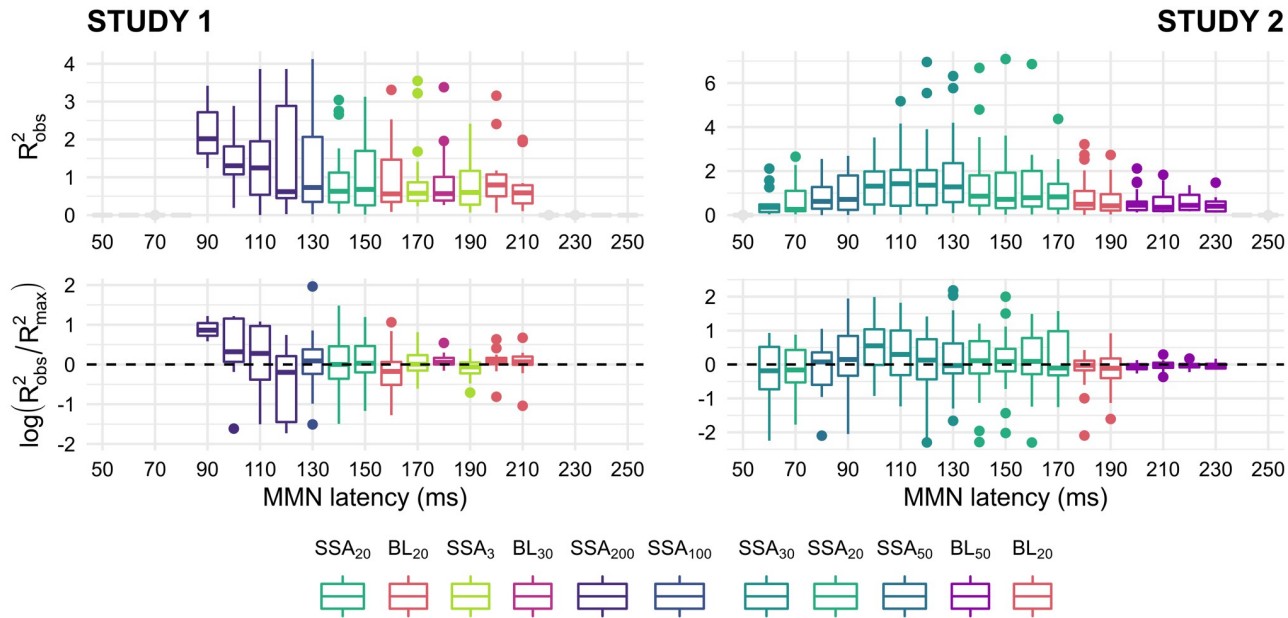

**Fig 10. Variance accounted for by best models.** (Top) Group distribution of the proportion of variance accounted for (denoted $R^2_{obs}$) by the best model family selected at each MMN latency by the Bayesian model comparison (see Fig 8) and for the parameter $\tau$ closest to the value estimated by Bayesian model averaging (see Fig 9). (Bottom) Comparison between the actual values of $R^2_{obs}$ and theoretical maximum proportions of variance that can be accounted for by the models ($R^2_{max}$), given the variance observed in the data. The ratios being positively skewed, they were log-transformed for normality (the optimal ratio value of 1 is thus transformed to 0).

## Discussion

In this paper we have revisited the trial-by-trial computational modeling of EEG-recorded scalp potentials in response to sounds embedded in oddball sequences. The study design focused on four aspects that have emerged in recent years as important for drawing reliable conclusions from computational studies. First, we specified the model space so as to represent all mechanisms that have been proposed for the generation of the MMN. Thus, models that were tested implemented both physiologically–inspired, adaptation–like dynamics and computational, mechanistic models of Bayesian statistical learning. For the latter family of models, different quantities were tested and compared following previous modeling work [76]: precision-weighted prediction error, Shannon's surprise, and model adjustment. Second, we used simulations to assess models discrimination and guide the interpretation of the findings obtained from the electrophysiological data. Third, we modeled the neural responses in a time-resolved fashion that allowed for latency-specific inferences as the latest research in animals suggests that MMN-like components might result from different mechanisms emerging along the time course of the evoked response and across the auditory hierarchy. Fourth, we provided our own replication by analyzing two independent datasets concurrently, which allowed us to focus on the findings common across the two datasets as the most likely to replicate.

As a discussion we will first synthesize the modeling results and speculate on the parallels that may be drawn from the findings in human and animal studies. Then, the significance and impact of methodological choices inherent to computational modeling will be discussed, highlighting the improvements we brought forth in the present study. We will conclude by

highlighting the potential limitations hindering our results and suggest improvements for future studies.

## Multiple mechanisms may underlie the MMN response at different latencies

Our results confirm a well-established fact in the MMN literature, namely that neural responses to individual stimuli embedded in a sequence reflect ongoing statistical learning rather than mere processing of physical properties and frequency of occurrence [93–96]. Here this is evidenced by the fact that the deviance detection model—a model capable of producing an MMN–like response but without any dependence on stimulus history—turned out to fit the data less well than all other history–sensitive models.

Compared to previous computational modeling studies on time-averaged data [76] or with restricted model space [71, 72, 75, 97], our approach allowed us to identify interesting temporal dynamics in generative models of the MMN. Family-wise model comparison at the group level identified a predominance of adaptation models (SSA) in both EEG datasets during the early MMN window from 80–90 to 150–170ms (Fig 7, BF >100 at most latencies). Importantly this finding was confirmed when gradually filtering out low SNR datasets (Fig 8), and could not be attributed to confusion of BL models with SSA as the latter model family showed good specificity throughout these early latencies, at least for Study 2 (Fig 6 lower panel, Study 2: $PPV_{SSA} > 0.95$). The model adjustment version of Bayesian learning—$BL^{madj}$ which indexed the MMN as the amount of updating between prior and posterior belief on the probability to get a deviant—then competed with SSA for about 20ms with slightly different latencies depending on the datasets (160–170ms for Study 1 and 170–180 for Study 2, Fig 7). Surprisingly, in high SNR samples the evidence for $BL^{madj}$ decreased in Study 1 but was reinforced in Study 2 (Fig 8). This discrepancy could potentially be explained by the lower specificity of SSA models in Study 1 compared to Study 2 within these time windows (Fig 6). Finally the Bayesian learning implementation of Shannon's surprise ($BL^{surp}$) showed the highest—but relatively moderate (BF >3)—evidence against competitors in the last part of the MMN response up to 230ms, a result made clearer and coherent across studies within the highest SNR samples (Fig 8, middle and lower panels). Yet $BL^{surp}$ and $BL^{pwpe}$ are not easily disambiguated under our experimental design as their predicted responses are highly correlated (r >0.99) and they consistently showed low specificity (PPV <0.95) or sensitivity (TPR <0.80) in time-resolved model recovery (Fig 6). Therefore, strict conclusions concerning the domination of $BL^{surp}$ in the late MMN response would be premature, our data being compatible with both surprisal and precision-weighted prediction errors signals.

To our knowledge this is the first modeling study of the MMN providing comparable results across two independent EEG datasets and on a substantial total sample of 82 participants. Dynamic Causal Modeling (DCM) studies on EEG data prompted early on towards a multiplicity of mechanisms supporting deviance detection along the cortical hierarchy [52]. Following this intuition Lieder et al. (2013)—in a pioneering computational modeling work—compared most of the potential mechanisms supporting the MMN on trial-wise scalp averaged EEG data [76]. However they could only draw strong conclusions at the level of superfamilies of models, pitting all BL models against phenomenological models (a grouping of adaptation and change detection models). In more fine-grained comparisons the model adjustment Bayesian learning model had the highest posterior probability but the evidence was not compelling. We can only speculate as to why no clear winner stood out in this initial study, but the low sample size (8 participants) and the temporal averaging of the signal seem to be plausible contributing factors in light of our own results. However these factors can hardly explain the

very low evidence for adaptation models in their study, which constitutes a major discrepancy with our results. Differences in designs and models implementations may have a significant impact, an issue that we will further discuss below. Differently, a recent computational modeling study of the somatosensory mismatch revealed partial evidence for a model adjustment BL process (there called "Bayesian surprise") from 150 to 200 ms both at scalp and source levels [97], reminiscent of our own results, although they did not test any adaptation–like models. Instead, at earlier latencies their data were better explained by a "confidence corrected surprise" BL model that we did not include in our analysis. We leave it for future studies to disambiguate between adaptation and BL "confidence corrected surprise" during the early window of the MMN. It is worthwhile to note that the same authors further showed the involvement of Bayesian learning mechanisms in the generation of the MMN for three different sensory modalities (auditory, somatosensory and visual), as well as a signature of cross-modal processing in a later mismatch component (P3a, 300–350 ms) [75].

To summarize, our results suggest the existence of distinct and potentially overlapping generative mechanisms of the scalp MMN following a temporal pattern. Adaptation–like response seems to dominate most of the MMN window from 80–90 to 150–170 ms, while BL–like processes appear during the late MMN with varying temporality depending on the paradigm (Study 1 or 2) (Fig 7). This temporal pattern may be explained in two distinct interpretation frameworks. One simple and tempting approach is to map ERP latencies on the cortical hierarchy, with early and late post-stimulus activations reflecting low- and high-level processing, respectively. In the present context that would lead us to attribute adaptation–like processes to sources of the MMN located in the temporal cortex and later BL–like responses to frontal areas. But the mixed results from MMN source localization studies does not quite support such a pattern. Frontal sources of the MMN were initially found to be be activated slightly later (8ms) than temporal ones in EEG [98], and correlation analyses of ERP amplitude with fMRI activations associate early MMN response (90–120ms) to signal change in right temporal cortex and late MMN response (140–170ms) with change in right frontal cortex [99]. However more recent studies provided evidence for the involvement of temporal and frontal sources at similar times. For example Philips et al. (2015) did not observe significant differences in the MMN peak latency between primary auditory cortex (A1), superior temporal gyrus (STG) and inferior frontal gyrus (IFG) for multiple deviance types (duration, frequency, gap, intensity and location) in MEG [56]. Another study found activations for frequency and intensity deviances in both supratemporal (Heschl's gyrus or planum polare depending on deviance type) and infrafrontal cortices (IFG) at the peak of the MMN (150–250ms), but also during early deviance (15–75ms) and the "rising edge" of the MMN (110–150ms, called elsewhere early MMN) on fused EEG-MEG data [58]. Finally generators for a frequency MMN were found in both A1 and IFG in EEG [100].

Rather than treating the time course of the ERP as reflecting a unidirectional stream of bottom-up cortical activation, an alternative approach consists in interpreting the modulations of the EEG signal as markers of dynamic changes in forward and backward transmission of information within the hierarchical temporo-frontal network generating the MMN. Inferring such temporal causal links require both source-level signals and computational models, such as in Dynamic Causal Modeling (DCM). To our knowledge, only one DCM study has looked into the temporal dynamics of such changes in causal connections. In one of their first DCM study, Garrido et al. evaluated the importance of backward connections for eliciting the ERP to deviants in an oddball task [50]. Their results clearly showed that if forward connections are necessary to explain the evoked response throughout the whole duration of the ERP, backward connections (between A1, STG and left IFG) have a particular influence later in time, starting around 200ms. Thus, the appearance of Bayesian learning–like mechanisms at late latencies in

our results could reflect the increased involvement of such backward connections, for example by casting newly updated predictions downward in the cortical hierarchy. Yet such interpretation is highly speculative and would need further fine–grained time–resolved DCM analyses to be verified in the future.

## Relationship to results from animal studies

Recent reviews—combined with novel empirical work—confirmed a high degree of homology between human MMN and rodent mismatch responses [101–103], justifying the use of translational research to better characterize the neurocognitive mechanisms of auditory perception. The gradient of generative mechanisms revealed by the present computational study is strongly reminiscent of electrophysiological results obtained in rats. Using control stimulus sequences and single-unit recordings in the rat auditory areas, Parras et al. [63] were able to decompose neuronal mismatch responses to oddball sequences into repetition suppression (RS) and prediction error (PE) signals. They showed that mismatch responses in the auditory cortex (AC) and in subcortical auditory areas reflected mainly repetition suppression, especially at latencies below 50ms—with PE signals emerging only later, around 75ms post-stimulus. Applying a similar methodology to the rat medial prefrontal cortex (mPFC), the same team later demonstrated that, in contrast to what has been reported in AC, mPFC mismatch responses are largely driven by PE signals [64]. However, because these frontal responses did not start until 200ms and lasted several hundred milliseconds, they may be analogues of human P3–like rather than MMN responses. More generally, the parallels between previously cited animal findings and ours should be treated with caution as the experimental paradigms differ sensibly in design and measures. Nevertheless, the temporal dynamics of physiological processes described in Parras et al. (2017) appear consistent with the timing of the gradient of computational models we report here, providing one takes into consideration that the latencies of sensory ERP components in rats are roughly twice as short than their human counterparts in comparable settings [104, 105]).

Our study identified a second type of gradient, this one pertaining to adaptation time constants of SSA models (Fig 9). Models with higher $\tau_a$, indicative of slow–adaptation (see Eq 6), dominated the onset of the MMN response and were progressively replaced by fast-adaptation models (lower $\tau_a$). Said differently, our modeling results suggest that the rate of adaptation to repeated auditory stimuli is not fixed throughout the entire duration of the neural response, but instead increases gradually. Although this gradient was present to some degree in both datasets, it was more pronounced in Study 1, especially due to very slow adaptation at the onset latency of the N1. One explanation for this discrepancy could be the variable ISI in Study 1, as previous studies suggest that temporal predictability enhances repetition suppression [106, 107].

Once again such results obtained at the computational level with human scalp EEG data are consistent with previous work at the electrophysiological level on auditory neurons of anesthetized rats [29]. Nieto-Diego and Malmierca (2016) first showed that in lemniscal auditory cortices (primary auditory cortex A1, anterior auditory field AAF and ventral auditory field VAF), SSA was significantly stronger during the late window of the rat MMN-like response (figure 5 from [29]). In contrast, SSA in non-lemniscal regions (posterior auditory field PAF and suprarhinal auditory field SRAF) did not change through time after sound onset but was stronger than in lemniscal regions until about 100 ms when adaptation in all auditory areas reached similar high levels (figure 5 from [29]). Additionally, they could estimate the rate of adaptation of individual neurons to standard tones and found adaptation to be faster for non-lemniscal regions (figure 7 from [29]). Finally, the negative component of the difference wave

of local-field potentials (LFP)—supposedly more representative of neuronal population processes recorded through non-invasive methods—peaks later in non-primary than primary cortical fields (figure 8 from [29]). To summarize their results, multiple lines of evidence seem to converge towards an increase of SSA in auditory cortex within the time-span of MMN-like neuronal responses. Yet the fact that adaptation rates are extremely fast in rats (in the order of one second) and estimated from neuronal firing rates makes it unclear whether similar physiological mechanisms underlie the gradient of adaptation that emerged from our modeling study.

## Conclusions from modeling studies critically depend on modeling choices and methodology

The value of the discussion so far rests upon the reliability of the results, i.e. on the ability of the computational modeling and inferential procedure (model selection) to capture some aspects of the generative mechanisms of the MMN. For the present study we have been careful to follow the most recent guidelines for computational modeling of neurophysiological data [91], notably by carrying out model recovery analyses which have largely been missing in previous computational studies of the MMN (at the exception of two recent examples [72, 97]). Our simulations showed that discriminating between models standing for candidate theories of the MMN was not straightforward—at least using classical oddball paradigms. Statistically powerful and reliable model inference required optimal conditions, including good signal quality (SNR; Fig 4) and sufficient sample size (Fig 5). The effects of sub-optimal conditions were complex and not entirely intuitive for whom is accustomed to more conventional analyses of EEG signal. In the classical frequentist analysis of ERP, low SNR is a well recognized issue that threatens statistical power—a serious brake to scientific discovery but one that can be mechanically compensated by increasing the sample size [108, 109]. Contrasting with this view, we have found that the candidate models got increasingly confounded when SNR decreased in a strongly biased way that favored some models in all circumstances, including when a competitor was the true generative model, thereby distorting the results rather than simply lowering the chance of positive findings (Fig 4). Unfortunately, the same simulations showed that the distortions introduced by poor signal quality could hardly be offset by the use of larger datasets, as increasing the sample size had primarily the effect of sharpening the confusion matrix but not so much of eliminating bias (said otherwise, large samples resulted in the posterior mass distribution being concentrated on a single model but not necessarily the true one, see Fig 5). This suggests that selecting participants with high quality data, even though it reduces the sample size, might be a worthy strategy to improve the reliability of model selection (Fig 8). However it should be noted that, even under the best conditions afforded by electrophysiological recordings, some models could no be disambiguated, the best example being the Shannon surprise and precision-weighted prediction error outputs of the Bayesian learning process. Altogether our simulations highlight the utmost importance of conducting model recovery analyses prior or jointly with Bayesian model inference, and to use the results of the simulations to inform the interpretation of model selection outcomes.

Besides data properties, analytical choices can also influence scientific conclusions, and it is worth reviewing some of the specific ways in which the dependence of results on methods manifests in computational modeling studies—especially regarding model comparison and selection.

The first consideration is the model space. By nature, BMC may pick a model out based on its performance in predicting data, but only relatively to other models. The evidence provided by BMC in favor of a model is only as good as the choice of the unfortunate competitors, and

the contribution of a modeling study to theoretical debates depends critically on how adequately candidate theories were translated in computational models. Recent unifying efforts have attempted to subsume adaptation dynamics under the predictive coding framework by reframing adaptation processes as a natural component of the predictive processing machinery, namely the attenuation of responsiveness of sensory cortices under the influence of top-down predictions [16, 110, 111]. The models implemented in the present study have very distinct origins: SSA as exponential decay and recovery is essentially a phenomenological function empirically derived from physiological observations, while Bayesian models are mathematical implementations of computational learning principles. This does not mean that they can not or need not be compared. On the contrary, the combined use of oddball and control stimuli sequences in animal studies allowed for the complete separation of repetition suppression and prediction error signals in single-cell recordings, suggesting distinctive mechanisms at the physical level [16, 63, 64, 112]. In addition, while predictive theories of the MMN interpret the electrophysiological component as an index of Bayesian inference on sensory input, they remain largely uncertain about which aspect of the Bayesian process is most reflected in the scalp field potential (prediction error, surprise, model updating, etc.). We have accommodated this uncertainty by declining the BL models in multiple versions each implementing a measure or output of the Bayesian learning process.

A second important consideration arises at the stage of group-level inference. There are two procedures available for Bayesian model comparison, each suited to distinct scenarios: fixed-effect analysis (ffx–BMC)—perfectly adapted to a situation where the same mechanism is to be found across all participants—and a random effect approach more appropriate to heterogeneous populations (rfx–BMC; [89, 113]). The latter is mathematically more sophisticated and as such have benefited from more methodological attention. It may even have been credited with more validity in the domain of brain data modeling, yet the authors of the rfx–BMC themselves present the two approaches as diverging primarily on the specific and distinct assumptions they make on the population structure. For the computational modeling of the MMN we have opted for a fixed-effect approach and this decision is more than a mere methodological concern. The selection of the most plausible population structure is a statement of theoretical interest in itself. While the homogeneous assumption suggests invariance, pointing towards a hard-coded, defining feature of the organism, heterogeneity implies inherent diversity and possibly plasticity. Therefore, as far as elucidating the mechanisms of a cognitive or neural process is concerned, the implications of deciding between either population structures are far-reaching. One epistemological implication is the reasonable expectation one may have about the contribution of modeling studies to debates around competing mechanistic hypotheses. Such contention can, at least in theory, be settled under the homogeneity assumption and optimal experimental conditions But this is not necessarily the case when ties in the competition are tolerated, as with a random-effect approach.

The possibility of multiple coexisting models is also conditioned by the way the temporal dimension of the targeted neural response is handled. There is a long history of debate around the number of cortical generators underlying the MMN [98, 114, 115] but most computational modeling studies of the MMN have treated it as a uniform component by averaging the signal of a large time-window in the spirit of classical ERP analysis. Yet, following the perspective offered by DCM studies [52, 57, 58], the MMN appears as a neural marker of a distributed, hierarchical, prediction (error)–based processing of internal and external information. In this view the electrophysiological signal is a mixture of multiple responses from the various units composing the predictive coding architecture, each with its own distinct dynamics, and we may anticipate that temporal averaging will blur the distinctions between them. In this perspective, time-resolved approaches as has been implemented here and in previous works [72,

73, 75, 97], appear absolutely essential for computational modeling studies to help elucidate the MMN mechanisms.

## Perspectives

The present paper has deployed and improved upon a computational approach to the dynamic modeling of brain responses to sequences of sounds, in the context of existing data collected from a classical oddball paradigm. Our results, suggesting separable underlying contributions to the MMN at differing temporal intervals, may have important implications for studies examining individual differences in MMN, for example in a clinical context, and may provide a mechanistic understanding of the nature of such differences. It is noteworthy that robust inference about the mechanisms of the auditory MMN could be drawn from such a simple paradigm, and future research should question whether the findings would replicate in more sophisticated designs including e.g. roving deviants [71, 76], predictable sequences [72], local-global violation rules or omission effects [68]. Another important question is what are the algorithmic underpinnings of the MMN at various latencies in cases where the adaptation mechanism is unlikely to play a significant role, e.g. for deviance features such as delay, duration or intensity or in the context of abstract rules. We would also like to acknowledge some limitations of the current study. First, the EEG datasets we used from previous publications were submitted to rather aggressive high–pass filtering settings. As explained in the Methods section, the use of a 2-Hz high-pass filter in the second study was necessary to remove low frequency artifacts caused by threat-induction. For both studies however, in the context of trial-wise modeling, strong high-pass filtering also permitted to remove signal drifts in the spectral range of interstimulus interval and below (0.5–1 Hz). Besides, our oddball paradigm being passive, no P3 component was elicited, which considerably reduces the risk of distortions of earlier components by zero-phase (non-causal) filters as previously shown with high-pass at 1.5-Hz and above [116]. Nevertheless, we would advise future replications to use lower high-pass filtering thresholds, when possible, as recommended by recent guidelines on the topic [116, 117]. Additionally, we could not implement cortically resolved modeling in this study given the EEG datasets we used were not optimized for source reconstruction analysis. As a final note, we encourage future work to refine the mechanistic accounts of the MMN by fitting time-resolved models on data with high spatial resolution such as MEG- or fused EEG/MEG-derived cortical sources [72], as well as by testing models that are more biophysically plausible [37, 65–68].

## Supporting information

**S1 Fig. Frontal ROIs selected for EEG data for both studies. A:** Study 1 EGI GSN200 system 128 electrodes layout, plotted on a head template. The frontal ROI is highlighted, comprising twelve frontal electrodes, including Fz. **B:** Study 2 Biosemi Active Two system 64 electrodes layout. For this study the selected ROI comprised only five frontal electrodes, also including Fz. (TIF)

**S2 Fig. Automatic detection of MMN peak for Study 1.** A Gaussian function has been fitted on each participant's average ERP at the selected ROIs. Participants for whom the gaussian fit has failed to result in a negative peak (due to an absence of detectable MMN, or presence of a positive deflection instead of the expected negativity) have been excluded from subsequent analyses. (TIF)

**S3 Fig. Automatic detection of MMN peak for Study 2.** Same method as for Study 1. (TIF)

**S4 Fig. Correlations between models used for the recovery analysis.** *SSA*: stimulus-specific adaptation, *BL*: Bayesian learning, *surp*: Shannon's surprise, *pwpe*: precision-weighted prediction error, *madj*: model adjustment.
(TIF)

**S5 Fig. Comparing assumptions about the distribution of models frequencies in the population.** Model evidence for the two population structures underlying the ffx-BMC (homogeneous population, $M_{FFX}$) and the rfx-BMC (heterogeneous population, $M_{RFX}$) have been calculated and contrasted latency-wise at the level of model families. Only participants who displayed a negative MMN at the target latency were included. The homogeneous population distribution ($M_{FFX}$) dominated over the entire MMN time-window in Study 2, and at most latencies in Study 1—except between 130 and 140ms. A closer inspection of models' estimated frequencies at t = 130ms and 140ms in Study 1 indicated that the slight evidence in favor of population heterogeneity at these latencies is driven by $BL^{madj}$ models competing with SSA models. Given that $BL^{madj}$ models become dominant in the whole sample at 160ms, the apparent heterogeneity 20ms earlier might be attributed to inter-individual variation in the time at which the dominant generative mechanism of the MMN switches between adaptation and Bayesian-like processes.
(TIF)

**S1 Text. Bayesian modeling of the population structure.**
(PDF)

## Acknowledgments

We would like to thank Kristien Aarts, Camille Fakche and Arthur Saverot for their help in collecting data for Study 2. We gratefully acknowledge support from the CNRS/IN2P3 Computing Center (Lyon—France) for providing computing and data-processing resources needed for this work.

## Author Contributions

**Conceptualization:** Arnaud Poublan-Couzardot, Françoise Lecaignard, Jérémie Mattout, Antoine Lutz, Oussama Abdoun.

**Data curation:** Arnaud Poublan-Couzardot, Enrico Fucci, Antoine Lutz, Oussama Abdoun.

**Formal analysis:** Arnaud Poublan-Couzardot, Oussama Abdoun.

**Funding acquisition:** Richard J. Davidson, Jérémie Mattout, Antoine Lutz, Oussama Abdoun.

**Investigation:** Enrico Fucci, Antoine Lutz.

**Methodology:** Arnaud Poublan-Couzardot, Françoise Lecaignard, Jérémie Mattout, Oussama Abdoun.

**Supervision:** Antoine Lutz, Oussama Abdoun.

**Visualization:** Oussama Abdoun.

**Writing – original draft:** Arnaud Poublan-Couzardot, Oussama Abdoun.

**Writing – review & editing:** Arnaud Poublan-Couzardot, Françoise Lecaignard, Enrico Fucci, Richard J. Davidson, Jérémie Mattout, Antoine Lutz, Oussama Abdoun.

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
