## [Decision Letter · Decision Letter 0]

27 Mar 2023

Dear Mr Poublan-couzardot,

Thank you very much for submitting your manuscript "Time-resolved dynamic computational modeling of human EEG recordings reveals gradients of generative mechanisms for the MMN response" for consideration at PLOS Computational Biology.

As with all papers reviewed by the journal, your manuscript was reviewed by members of the editorial board and by several independent reviewers. In light of the reviews (below this email), we would like to invite the resubmission of a significantly-revised version that takes into account the reviewers' comments.

First of all, my apologies for the long time it took to get back to you--first it was a major challenge to find editors for this work (I as a section editor have taken this over once the editor who was assigned after a long time dropped it again), and then it was a struggle to find reviewers. Nevertheless, all reviewers are quite positive about the manuscript, requesting mostly clarifications and assessments of the generality of your results. I am nevertheless giving you a major revision so you have more time to complete it.

We cannot make any decision about publication until we have seen the revised manuscript and your response to the reviewers' comments. Your revised manuscript is also likely to be sent to reviewers for further evaluation.

Sincerely,

Marieke Karlijn van Vugt, PhD

Section Editor

PLOS Computational Biology

Marieke van Vugt

Section Editor

PLOS Computational Biology

First of all, my apologies for the long time it took to get back to you--first it was a major challenge to find editors for this work (I as a section editor have taken this over once the editor who was assigned after a long time dropped it again), and then it was a struggle to find reviewers. Nevertheless, all reviewers are quite positive about the manuscript, requesting mostly clarifications and assessments of the generality of your results. I am nevertheless giving you a major revision so you have more time to complete it.

Reviewer's Responses to Questions

**Comments to the Authors:**

Reviewer #1: This is an excellently written and informative paper on the putative computational mechanisms of mismatch negativity, a classical neural response which has been linked to adaptation, deviance detection, and Bayesian surprise/learning. The authors disambiguate between these different interpretations in a model-based time-resolved analysis of two large EEG datasets. The manuscript will be of interest for a large audience of cognitive, computational, and systems neuroscientists.

1. The introduction is very well written - however, I would encourage authors to also discuss more recent computational work on somatosensory and multimodal mismatch responses, which does differentiate between early and late electrophysiological signatures of surprise (dois: 10.1371/journal.pcbi.1008068 and 10.1101/2022.10.27.514010)

2. I think the use of the word “declined” in the sentence in line 104 is confusing; perhaps a better alternative would be “included”, “nuanced”, “disambiguated” etc.

3. Methods: The filtering settings (double high-pass filter, first at 0.5Hz, then at 1Hz) need to be revised and/or justified in light of previous analyses showing that such high cut-off frequencies introduce strong distortions in ERP amplitudes and latencies (e.g., doi: 10.3389/fpsyg.2012.00131) . This is even more problematic for study 2, where a 2-Hz high-pass filter was used.

4. The choice to only model ERPs in sensor-space is a limitation of the study. The study would benefit from a more direct analysis of separation between hierarchical levels and/or time scales of adaptation based on source reconstruction of ERPs. This is to some extent addressed in the discussion, but the datasets presented here yield a high potential to clarify some of the controversies regarding the source-level implementation of MMN present in the literature. I would encourage the authors to either include such an analysis in the paper, or discuss the reasons for not including it.

5. What was the reason for constructing several SSA models with fixed adaptation parameters rather than fitting these adaptation parameters to the data? Similarly, could forgetting constants be fitted to the data in case of BL models?

6. The text in line 275 mentions six SSA models but there are seven taus in line 276. Could the authors clarify this discrepancy?

7. Line 280, typo: “We inspired” - should be “We were inspired”?

8. Between lines 289-290, typo: “Ajustment” - should be “Adjustment”

9. While the model recovery analysis is a valuable part of the manuscript, I wasn’t convinced by limiting it to one set of taus, especially given the fact that the authors explicitly consider a wide range of fixed taus when fitting their models (see point 5). It would be good to see a more complete recovery analysis (with a range of taus), and/or a specific recovery analysis focusing on recovering parameters (taus) rather than models.

10. In discussing their absolute goodness of fit, the authors may want to refer to a paper (albeit not based on MMN) where similar R2 values have been found (doi: 10.7554/eLife.11476.001)

11. In discussing the possibility that different MMN latencies reflect different mechanisms, the authors may want to refer to a paper which showed that early latencies reflect deviance processing itself while late latencies reflect deviance feature processing (doi: 10.3389/fnhum.2021.613903), suggesting a difference in the levels of abstraction between early and late responses.

12. In discussing the work of Garrido in the context of backward connections, another piece of evidence comes from their 2009 paper (doi: 10.1016/j.neuroimage.2009.06.034), where adaptation has been linked to modulation of intrinsic connectivity (gain parameters) while model learning has been linked to modulation of extrinsic (e.g., backward) connectivity.

Reviewer #2: Poublan-Couzardot and co-authors tested in their manuscript entitled 'Time-resolved dynamic computational modeling of human EEG recordings reveals gradients of generative mechanisms for the MMN response’ different computational models and fitted these to EEG-Date to reveal the underlaying mechanism of the MMN. This is another important step in the attempt in recent years through model-based analysis techniques to better describe the dynamics of brain signals for mismatch responses. The manuscript is clearly written and the analyses and results are quite easy to follow. However, to enable other scientists to perform the analyses on their data as well, it would be nice if the authors described the individual steps with the VBA toolbox in more detail. The other point that is important to me is that the analysis should not only be performed on pre-selected electrodes. I think it is important to show not only the temporal effects but also the spatial specification of the results since it is conceivable that different models are dominating over different brain regions. I know that this is computationally expensive and that the results are not easy to present, but it is possible and would add significant scientific value to the results from my point of view (please see for a similar approach Gijsen et al., PloS Comput Biol. 2021). If the manuscript becomes too long due to the additional analysis of the whole electrode space, from my point of view the simulations could be taken into the supplement.

Minor:

The title should better reflect the results of the study.

It was unclear to me why the model was not fitted until 45 ms. One would certainly not expect large effects in this time, but it would contribute to the completeness of the analysis.

The average over 10 ms also seems quite long to me. Despite the additional computing time, the authors should consider increasing the temporal resolution, e.g. 5 ms.

Reviewer #3: This paper details a very thorough and yet accessible approach to assessing which of a series of models can best account for changed responsiveness to auditory stimuli in an oddball sequence. The results show quite convincingly that different processes are likely to be most influential at different time points within the auditory event-related potential.

The paper is exquisitely written. The material is complex for a reader from a non-computational background but the way it is presented, and the thoroughness of the approach, makes this a very informative and convincing article. I have no major issues and only a few very minor comments that the authors might consider.

p20 line 302. Is the prior mean you refer to here implemented at an individual level?

Figure 2 - although in text you say there isn't much to glean from the BL measures it does seem that where there is a tendency to outperform others it seems always to be 20-50ms. Is this worth noting?

I think the data displays you have chosen are quite spectacular in communicating the outcomes of the analyses. It was a pleasure to read this paper and I believe it will be an excellent contribution to the literature.

**Have the authors made all data and (if applicable) computational code underlying the findings in their manuscript fully available?**

Reviewer #1: None

Reviewer #2: Yes

Reviewer #3: Yes

PLOS authors have the option to publish the peer review history of their article (what does this mean?). If published, this will include your full peer review and any attached files.

Reviewer #1: No

Reviewer #2: No

Reviewer #3: No
---

## [Decision Letter · Decision Letter 1]

20 Nov 2023

Dear Mr Poublan-couzardot,

We are pleased to inform you that your manuscript 'Time-resolved dynamic computational modeling of human EEG recordings reveals gradients of generative mechanisms for the MMN response' has been provisionally accepted for publication in PLOS Computational Biology.

Best regards,

Marieke Karlijn van Vugt, PhD

Section Editor

PLOS Computational Biology

Marieke van Vugt

Section Editor

PLOS Computational Biology

I decided to accept this revision, even though one of the reviewers still has comments. I think that adding a spatial component to the MMN generation will make this already complex paper difficult to follow, and the other two reviewers are happy with your work. Congratulations!

Reviewer's Responses to Questions

**Comments to the Authors:**

Reviewer #2: It is very unfortunate that the authors only address the reviewers' points through argumentation and not through additional data analysis. From my point of view, there is no apparent reason not to examine the spatial distribution of the models as well. Analyzing the data based on signals from ROIs, which also include different electrodes and which are motivated based on previous publications (which I unfortunately could not access) seems insufficient from my point of view. I also did not find it necessary to go into the source space. Just showing the topology and in which electrodes which models describe the data is extremely informative. Therefore, from my point of view, it is necessary to do this analysis. This seems to me also clearly more important than the model recovery part, which can be taken over into the supporting information.

From my point of view, it is necessary to show the spatial distribution of the models across the electrodes for several reasons:

1. a plausible distribution corresponding, for example, to the electrodes from which the present signal was averaged shows that the model(s) are plausible.

2. signal from electrodes that do not explain any model serve as natural reasonable controls (over for example occipital regions).

3. It is conceivable that different models can explain the MMN signals not only in time but also for different brain areas. Inferring that this is not the case because auditory MMN is quite homogeneous may be true, but should be tested.

Reviewer #3: Thank you for addressing my comments. I am very happy with the revised version.

**Have the authors made all data and (if applicable) computational code underlying the findings in their manuscript fully available?**

Reviewer #2: None

Reviewer #3: Yes

PLOS authors have the option to publish the peer review history of their article (what does this mean?). If published, this will include your full peer review and any attached files.

Reviewer #2: No

Reviewer #3: No

---

## [Editor Report · Acceptance letter]

7 Dec 2023

PCOMPBIOL-D-22-01345R1 

Time-resolved dynamic computational modeling of human EEG recordings reveals gradients of generative mechanisms for the MMN response

Dear Dr Poublan-couzardot,

I am pleased to inform you that your manuscript has been formally accepted for publication in PLOS Computational Biology. Your manuscript is now with our production department and you will be notified of the publication date in due course.

With kind regards,

Zsofi Zombor
